# [18F]F-AraG imaging reveals association between neuroinflammation and brown- and bone marrow adipose tissue
Jelena Levi [1] ✉, Caroline Guglielmetti[2,3,4], Timothy J. Henrich [5], John C. Yoon [6], Prafulla C. Gokhale [7], David A. Reardon [7], Juliet Packiasamy[1], Lyna Huynh[1], Hilda Cabrera[1], Marisa Ruzevich[1], Joseph Blecha[3], Michael J. Peluso [8], Tony L. Huynh [3], Sung-Min An[6], Mark Dornan[9], Anthony P. Belanger [9], Quang-Dé Nguyen[10], Youngho Seo [3], Hong Song[11], Myriam M. Chaumeil [2,3], Henry F. VanBrocklin[3] & Hee-Don Chae[1]

Brown and brown-like adipose tissues have attracted significant attention for their role in metabolism and therapeutic potential in diabetes and obesity. Despite compelling evidence of an interplay between adipocytes and lymphocytes, the involvement of these tissues in immune responses remains largely unexplored. This study explicates a newfound connection between neuroinflammation and brown- and bone marrow adipose tissue. Leveraging the use of [18F]F-AraG, a mitochondrial metabolic tracer capable of tracking activated lymphocytes and adipocytes simultaneously, we demonstrate, in models of glioblastoma and multiple sclerosis, the correlation between intracerebral immune infiltration and changes in brown- and bone marrow adipose tissue. Significantly, we show initial evidence that a neuroinflammation-adipose tissue link may also exist in humans. This study proposes the concept of an intricate immuno-neuro-adipose circuit, and highlights brown- and bone marrow adipose tissue as an intermediary in the communication between the immune and nervous systems. Understanding the interconnectedness within this circuitry may lead to advancements in the treatment and management of various conditions, including cancer, neurodegenerative diseases and metabolic disorders.

Adipose tissue (AT) is a heterogenous organ with a complex function crucial for diverse processes including energy storage, endocrine signaling and immunomodulation[1]. White and brown adipose tissues are the most extensively studied, but brite (brown-in-white)[2] and bone marrow (BM)[3] AT are attracting growing interest. The expansion of interest in brown and brown-like, brite, AT stems primarily from their therapeutic potential in diabetes and obesity, and association with cardiovascular health[4]. Rich in mitochondria, brown adipose tissue (BAT) can be activated by various stimuli, from cold exposure to adrenergic agonists, resulting in increased levels of lipolysis and glycolysis. These stimuli can also induce britening, a differentiation of white adipocytes into metabolically active brown-like

adipocytes, offering a potentially powerful strategy for weight loss and overall improvement of metabolism[5]. Despite strong evidence of a close bidirectional link between adipocytes and lymphocytes during immune responses[6,7], the immunomodulatory role of AT has also been primarily studied in the context of obesity with a focus on white AT. The connection between brown fat and immunity has not been studied extensively, but new studies provide evidence that batokines, signaling molecules secreted by BAT, affect not only metabolism but also systemic immune responses[8,9].

The involvement of BAT in modulating neuroinflammation has not been investigated since it was first demonstrated within the context of immune-neuro-endocrine communication by a Yugoslav immunologist,

[1]CellSight Technologies Incorporated, San Francisco, CA, USA. [2]Department of Physical Therapy and Rehabilitation Science, University of California San Francisco, San Francisco, CA, USA. [3]Department of Radiology and Biomedical Imaging, University of California San Francisco, San Francisco, CA, USA. [4]Mallinckrodt Institute of Radiology, Washington University in St. Louis, St. Louis, MO, USA. [5]Division of Experimental Medicine, University of California San Francisco, San Francisco, CA, USA. [6]Division of Endocrinology, Department of Internal Medicine, University of California Davis School of Medicine, Davis, CA, USA. [7]Dana-Farber Cancer Institute, Boston, MA, USA. [8]Division of HIV, ID and Global Medicine, University of California San Francisco, San Francisco, CA, USA. [9]Molecular Cancer Imaging Facility, Dana-Farber Cancer Institute, Boston, MA, USA. [10]Lurie Family Imaging Center, Dana-Farber Cancer Institute, Boston, MA, USA. [11]Department of Radiology, Stanford University, Palo Alto, CA, USA. ✉e-mail: jlevi@cellsighttech.com

Branislav Jankovic, in a series of studies in rats in the 1970s and 1980s[10–12]. Those studies showcased immunosuppressive quality of BAT by demonstrating that its removal leads to a more severe experimental autoimmune encephalomyelitis (EAE), a disease characterized by monocyte and T cell infiltration in the brain. Interestingly, the suppression of the immune system was noted only in animals whose BAT was removed during the neonatal phase or after cold acclimatization[10,12]. Given the critical role of BAT activation in maintaining body temperature in newborns and cold conditions[13], this indicates that the immunosuppression might be specifically related to activated BAT. Our study, for the first time to our knowledge, offers clear visualization of a link between neuroinflammation and BAT activation. Furthermore, we demonstrate that neuroinflammation co-occurs not only with BAT activation, but also with changes in bone marrow adipose tissue (BMAT). The observation of these phenomena was achieved by the use of a mitochondrial metabolic tracer, [¹⁸F]F-AraG (2'deoxy-2'[¹⁸F]Fluoro-9-β-D-arabinofuranosylguanine)[14], that has a distinctive ability to not only detect mitochondrial changes in activated T cells[15–18], but reveal increased mitochondrial biogenesis in AT as well. In this study, we first establish [¹⁸F]F-AraG's ability to visualize activated adipocytes by assessing its uptake in adrenergically stimulated brown fat and BM tissue. We then utilize this ability to simultaneously track activated lymphocytes and adipocytes to detect neuroinflammation-associated changes in BAT and BMAT in preclinical models of glioblastoma (GBM) and multiple sclerosis (MS). Finally, and most significantly, we present preliminary evidence that a neuroinflammation-adipose tissue link may also exist in humans. Our results support the existence of an immuno-neuro-adipose circuit, and implicate AT as one of the communication channels between the immune and nervous systems.

## Results

### [¹⁸F]F-AraG accumulates in BAT in response to adrenergic stimulation but not insulin administration

In mice, regulation of BAT by the sympathetic nervous system (SNS) is predominately driven by β3 adrenergic receptor signaling[19,20]. To examine [¹⁸F]F-AraG's accumulation in activated BAT, we treated mice with BRL37344, a well-studied β3 adrenergic agonist[21]. One time administration of BRL37344 led to a considerable accumulation of [¹⁸F]F-AraG in intrascapular BAT (iBAT) (Fig. 1a, b). Cold exposure also led to tracer accumulation in iBAT (Supplementary Fig. 1). Control animals, as well as animals that were treated with insulin, showed significantly lower [¹⁸F]F-AraG uptake in iBAT. In comparison, ¹⁸FDG, the tracer most commonly used for imaging BAT activation, showed an increased accumulation in iBAT not only with BRL37344 stimulation but in insulin-treated animals as well (Fig. 1c). These results suggest that, unlike ¹⁸FDG, [¹⁸F]F-AraG accumulates selectively in adrenergically stimulated iBAT.

Interestingly, chronic treatment with BRL37344 resulted in [¹⁸F]F-AraG accumulation not only in iBAT, but also in the axilla, lumbar region, and BM of the tibia and femur (Fig. 1d). In comparison to acutely treated mice, chronic adrenergic stimulation led to a comparable [¹⁸F]F-AraG accumulation in iBAT (Supplementary Fig. 2a), but significantly increased uptake in the vertebrae, especially in the lumbar region (Fig. 1d–g). The small size of mice presents a challenge in distinguishing the signal emanating from the vertebral BM from that originating in the spinal cord. Nonetheless, the segmentary pattern of uptake within the spine (Supplementary Fig. 2b), along with the absence of cellular changes in the spinal cord with BRL37344 treatment (Supplementary Fig. 2c, d), indicate that the signal detected in the vertebrae stems from the vertebral BM.

### Adrenergic stimulation increases adipocyte population in the brown and bone marrow adipose tissues

As a mitochondrial metabolic tracer[14,18,22] (Supplementary Fig. 3), [¹⁸F]F-AraG can be taken up by both stimulated T cells and adipocytes. To better understand the observed increased uptake in iBAT and BM, we utilized quantitative PCR and flow cytometry to examine changes that occur in

those tissues with chronic adrenergic stimulation (Supplementary Figs. 4–7, Supplementary Table 1, 2).

In iBAT, chronic BRL37344 treatment led to a dramatic increase in adipocytes (Fig. 2a), reported to occur through adrenergic signaling[23], along with expected morphological changes (Supplementary Fig. 4b) and upregulation of genes associated with activated brown fat and thermogenesis (Fig. 2b). BRL37344 treatment significantly increased the frequency of T cells (p = 0.006), with both CD4+ and CD8+ populations expanding (Fig. 2c). While the composition of the CD8+ subset did not change significantly after adrenergic stimulation (Supplementary Fig. 7a), the CD4+ population showed an increase in the frequency of naïve subset (p = 0.03) (Fig. 2d). Additional analyses of cellularity in different lineages and cell subsets are provided in Supplementary Fig. 7.

Upon chronic adrenergic stimulation, femoral BM, regarded representative of the vertebral BM as well[24], showed a significant increase in non-hematopoietic cells (p = 0.0006, Fig. 2e), and, interestingly, in the expression of a "briteness" marker Cidea[25] (p = 0.006, Fig. 2f). Given the critical role mitochondria play in [¹⁸F]F-AraG cellular accumulation, we analyzed changes in the mitochondrial content occurring in the femur BM after adrenergic stimulation (Fig. 2g). Treatment with adrenergic agonist resulted in close to two-fold increase in non-hematopoietic cells with high mitochondrial content, while hematopoietic cells showed no difference in mitochondrial staining. Most mitochondria-high/lineage-negative cells were positive for Nile Red, a stain for intracellular lipid droplets, indicating an adipocyte character for the mitochondria-rich cells (Fig. 2h). Nile Red-positive, lineage-negative cells constituted nearly the entirety of the mitochondria-high population, while T cells comprised only a marginal amount (Fig. 2i). To further characterize the mitochondria-rich, Nile Red-positive population, we evaluated their forward scatter intensity (FSC), a metric proportional to the cell's diameter. Consistent with previously documented large adipocyte size[26], the Nile Red-positive/lineage negative cells exhibited larger FSC values compared to other BM cells (Supplementary Fig. 7j).

Furthermore, neither the T cell frequency or the composition of the CD4+ population changed in BMAT after the BRL37344 treatment (Fig. 2j, k). Interestingly, femoral BM contained a higher frequency of CD4+ cells expressing tissue retention marker, CD69+[27] and activation/exhaustion marker PD-1+[28] (Fig. 2l). As adrenergic stimulation led to a decrease in mitochondrial content in T cells (Fig. 2m) indicating T cell dysfunction[29], the increased frequency of CD69+ and PD-1+ populations may reflect an exhausted, immunosuppressive microenvironment in the BMAT[30].

Overall, because in iBAT adrenergic stimulation led to the proliferation of both adipocytes and T cells, neither could be ruled out as the primary source of [¹⁸F]F-AraG accumulation. However, in the BM, flow cytometry showed an increase in mitochondria-rich adipocytes but no changes in T cells. Combined with the observed increase in expression of the briteness marker Cidea, results strongly suggests that brite-like BM adipocytes are the primary site of [¹⁸F]F-AraG accumulation in the adrenergically stimulated BM.

### [¹⁸F]F-AraG accumulates in adrenergically stimulated brown and bone marrow adipocytes

While various positron emission tomography (PET) tracers have been used for imaging activated BAT[31], PET imaging of metabolically active BMAT has not been reported to date[3]. To elucidate [¹⁸F]F-AraG's accumulation in BAT and BMAT and distinguish its uptake in T cells from adipocytes, we studied adrenergic stimulation in a mouse model with impaired thermogenesis in AT[32,33]. Recently reported Letmd1 Knock Out (KO) represents a suitable model for our studies because Letmd1 deficiency affects AT responsiveness to β3 adrenergic stimulation[33] without impacting immune cells (Supplementary Fig. 8a). In contrast to the wild type (wt) mice, chronic adrenergic stimulation of Letmd1-KO mice did not increase [¹⁸F]F-AraG uptake in either iBAT or lumbar, tibial and femoral BMAT (Fig. 3a, b), indicating that, within these tissues, the tracer accumulates primarily in stimulated adipocytes. The decreased mitochondrial content of the iBAT in

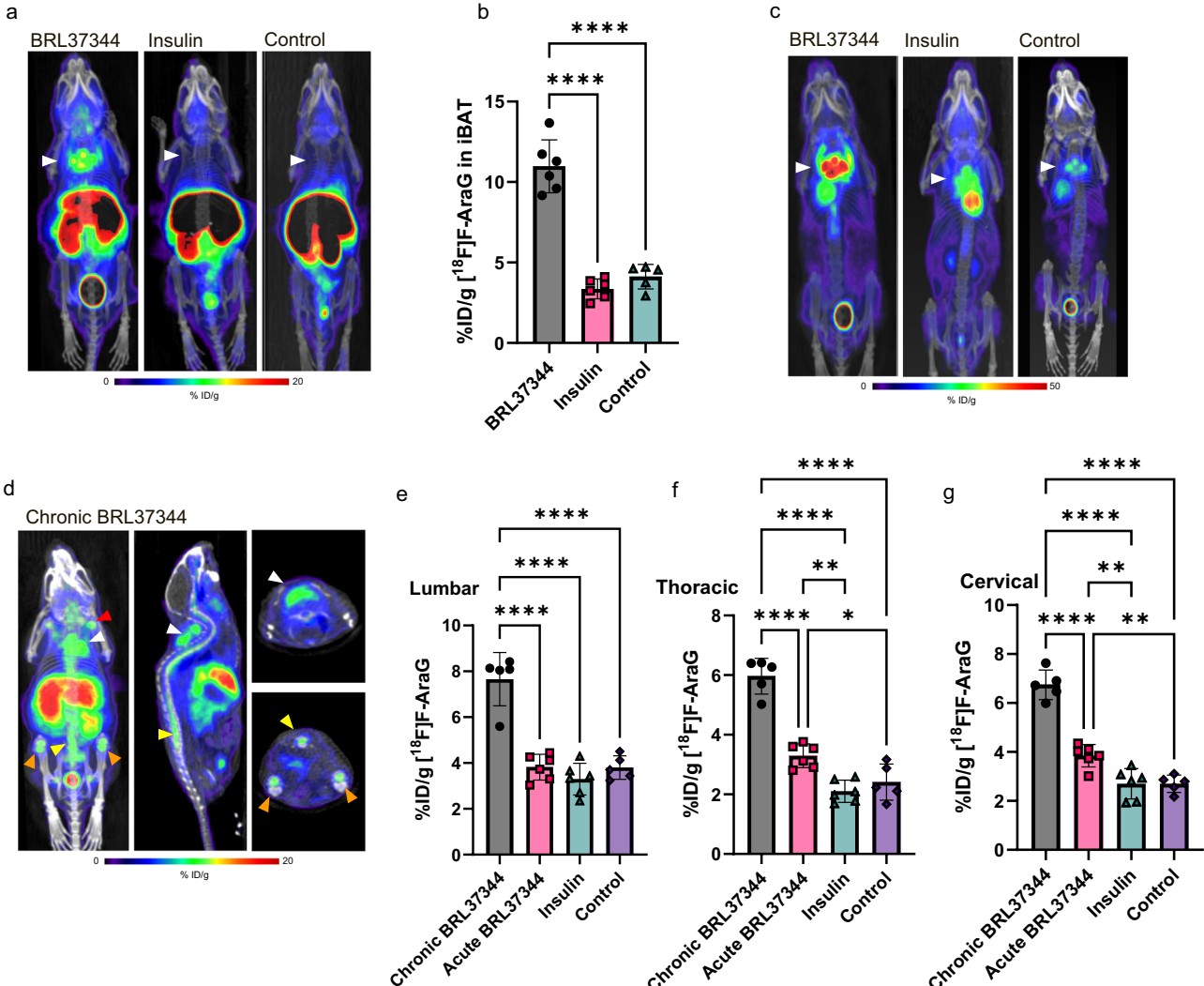

**Fig. 1 | [¹⁸F]F-AraG accumulates in adrenergically stimulated brown fat and bone marrow. a** Administration of β3 adrenergic receptor agonist BRL37344 (10 mg/kg) 1 h before imaging led to a high accumulation of [¹⁸F]F-AraG in iBAT (white arrowhead). No significant uptake was observed in the insulin-treated or control mice. **b** Signal in the iBAT of adrenergically stimulated mice (10.98 ± 1.64%ID/g) was significantly higher than the signal in the insulin treated (3.36 ± 0.62%ID/g) and control mice (4.11 ± 0.76%ID/g). No significant differences in iBAT signal were found between insulin-treated and control mice ($p = 0.53$). **c** The baseline ¹⁸FDG uptake in iBAT (white arrowheads) that was observed in control mice was increased in both insulin and BRL37344 treated mice, indicating lack of selectivity for adrenergic stimulation. **d** Administration of BRL37344 (10 mg/kg) for 4 consecutive days led to increased [¹⁸F]F-AraG signal in the intrascapular BAT (white arrowhead) and axillary BAT (red arrow) but also in the lumbar vertebrae region (yellow arrowhead), and the bone marrow of the tibia and femur (orange arrowhead). **e–g** Signal in the lumbar, thoracic, and cervical vertebrae of chronically stimulated mice was significantly different than the signal in acutely BRL37344- treated, insulin-treated and control mice. ID injected dose. [¹⁸F]F-AraG uptake in different region of interest was calculated as %ID/g. Data are plotted as mean ± SD ($n = 5$ or 6). Each spot represents an individual animal. $*p \leq 0.05$, $**p \leq 0.01$, $***p \leq 0.001$, $****p \leq 0.000$.

*Letmd1*-KO mice, along with its unresponsiveness to adrenergic stimuli[33], aligns well with our imaging findings. Likewise, Letmd1 deficiency desensitized BMAT to adrenergic activation. The expression of thermogenesis markers remained undetectable (Supplementary Fig. 8b) and there were no observable changes in either adipocytes or T cell population (Fig. 3c). Upon adrenergic stimulation, BM adipocytes, which generally exhibited a higher mitochondrial content compared to T cells (Fig. 3d), displayed a reduction in mitochondria-rich population ($p = 0.0009$, Fig. 3e).

To complement the studies in *Letmd1*-KO mice, we imaged adrenergic stimulation in T cell-deficient, *Rag1*-KO mice (Fig. 3f). Chronic BRL37344 treatment of T cell-deficient mice resulted in a signal increase in iBAT (Fig. 3g), indicating activated brown adipocytes as the primary target of [¹⁸F]F-AraG accumulation in this tissue. However, similar to *Letmd1*-KO, [¹⁸F]F-AraG signal in the vertebrae, femur, and tibia, was notably absent in adrenergically treated *Rag1*-KO mice, implying T cell involvement in the activation of BMAT. Following adrenergic stimulation, adipocyte population in

the BM increased (Fig. 3h). However, this increase did not correspond to the subset with high mitochondrial content (Fig. 3i, j), aligning with the reduced levels of Pgc1α, a critical regulator of mitochondrial biogenesis (Supplementary Fig. 8c).

The skeletal regions with high [¹⁸F]F-AraG accumulation in chronically stimulated mice closely match the areas containing regulated BMAT (rBMAT), a distinct type of BMAT[26]. Located in the red marrow, rBMAT is responsive to cold exposure, and, unlike BMAT at other skeletal sites, reactive to stimulation with β3 adrenergic agonists[34]. To confirm [¹⁸F]F-AraG's accumulation in β3-responsive BMAT, we imaged chronically treated wt mice with ¹⁸FDG. ¹⁸FDG accumulates in activated immune cells[35], and stimulated iBAT, but not in rBMAT[3]. In contrast to [¹⁸F]F-AraG (Fig. 1d), ¹⁸FDG uptake in the vertebrae, tibia and femur of chronically stimulated mice was minimal compared to its accumulation in iBAT (Fig. 3k). This result reinforces the findings in *Letmd1*-KO and *Rag1*-KO mice, providing additional evidence that the [¹⁸F]F-AraG signal in the BM

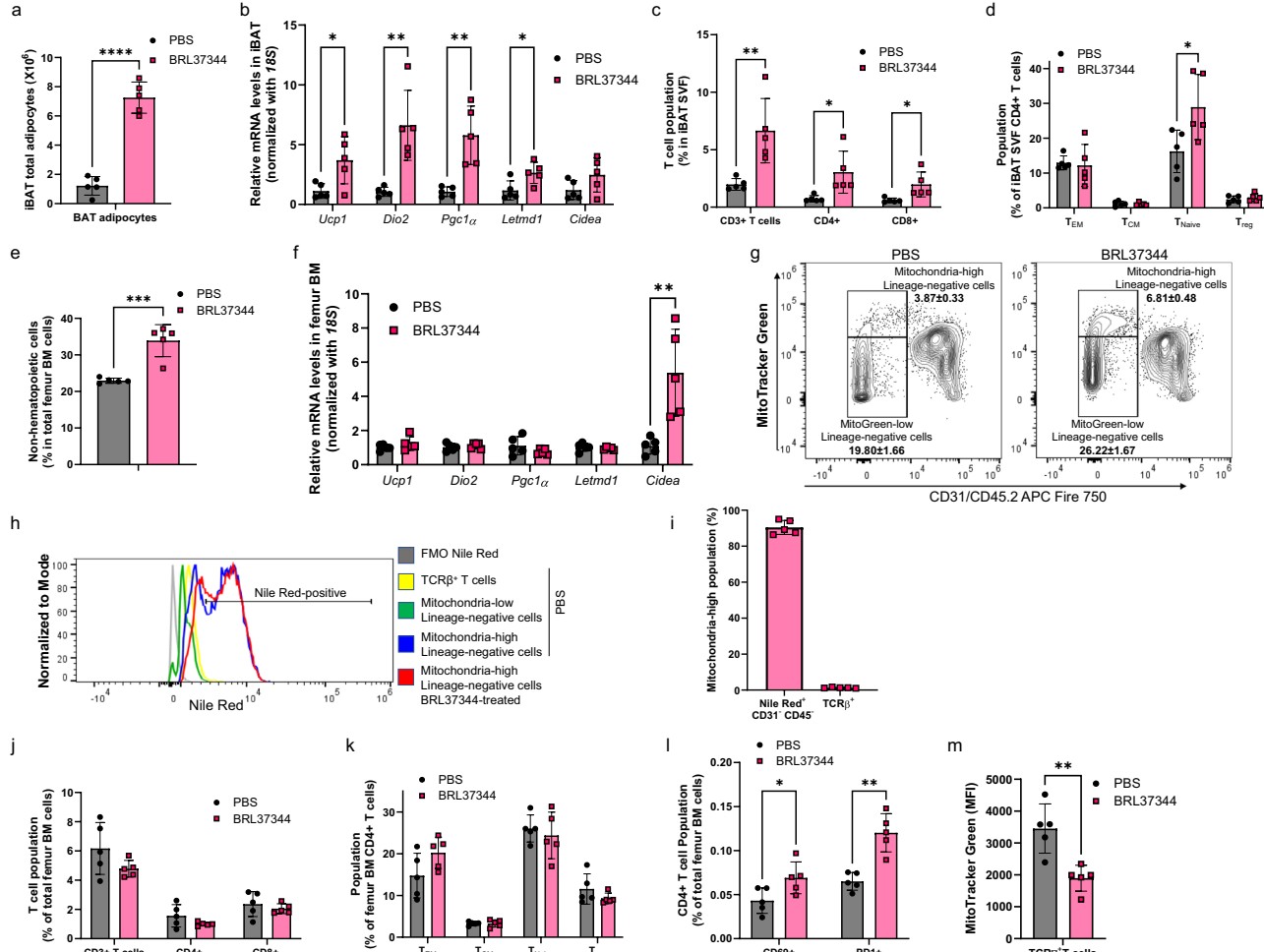

**Fig. 2 | Adipocytes are enriched in the brown and bone marrow adipose tissue of mice chronically treated with adrenergic agonist. a** Chronic adrenergic stimulation led to a 6-fold increase in the number of adipocytes in the iBAT. **b** Following treatment with adrenergic agonist, mRNA expression of markers of activation and thermogenesis were increased in iBAT. Expression of each gene was normalized against *18S* expression level. Relative expression levels are presented as fold induction over PBS control mice. **c** Increases in both CD4+ and CD8+ population contributed to the increase in T cells in iBAT post treatment with BRL37344. **d** Adrenergic agonist led to the change in subset composition of CD4+ T cells, with naïve cells increasing in frequency. **e** Following adrenergic stimulation, proportion of non-hematopoietic population increased in the femoral bone marrow. **f** Expression of *Cidea*, a briteness marker, increased in the bone marrow post adrenergic treatment, while the expression of other thermogenesis-related genes remained the same. **g** Representative flow cytometry plots show an increase in lineage-negative mitochondria-rich population in femur bone marrow following adrenergic stimulation. **h** Majority of mitochondria-rich cells were positive for lipid droplets staining with Nile Red dye. **i** Post adrenergic stimulation, overwhelming majority of mitochondria-rich population in the bone marrow contained lipid droplets, while T cells represented only a negligible portion. **j** Treatment with BRL37344 did not significantly change the proportion of T cells in femur bone marrow. **k** In contrast to iBAT, no significant changes were noted within the CD4+ population in the bone marrow after adrenergic stimulation. **l** In the bone marrow, frequency of CD4+ cells expressing retention marker CD69 and exhaustion/activation marker PD-1 increased post administration of the adrenergic agonist. **m** The graph shows median fluorescence intensities (MFI) of MitoTracker Green of femur bone marrow T cells. Following adrenergic stimulation, mitochondrial content of bone marrow T cells decreased ($p = 0.004$), suggesting immunosuppression. Data are shown as mean ± SD ($n = 5$). Each spot represents an individual mouse. * $p \leq 0.05$; ** $p \leq 0.01$; *** $p \leq 0.001$; **** $p < 0.0001$.

originates from its uptake in metabolically active BM adipocytes rather than its accumulation in T cells.

## GBM induced-neuroinflammation is associated with activation of BAT and BMAT

[18F]F-AraG's ability to track adrenergically stimulated adipocytes led to an unexpected discovery of AT activation in mice with bioluminescently-tagged syngeneic GL261 tumors. Given its minimal uptake in the healthy brain, we hypothesized that [18F]F-AraG could facilitate clear visualization of intracerebral inflammation, offering insights into immune responses during checkpoint inhibitor therapy. To understand the kinetics of anti-tumor immune response, we longitudinally imaged mice with intracranial GBM tumors during anti-PD-1/CTLA-4 therapy (Fig. 4a), tracking tumor growth with bioluminescent imaging. Surprisingly, during longitudinal

imaging, we observed activation of iBAT in half of the GBM affected mice (Fig. 4b, c). The activation, which was absent in subcutaneous tumor models (Supplementary Fig. 9a), was noted in both treated and untreated mice (Fig. 4b, d), but varied in timing and intensity among animals.

The most prominent iBAT activation was detected in an untreated mouse with the highest neuroinflammation as evidenced by the intracerebral [18F]F-AraG signal (M3, Fig. 4d). The iBAT [18F]F-AraG signal increased nearly ninefold as the tumor progressed over a span of 9 days. Notable [18F]F-AraG accumulation was observed in multiple BAT depots[36], as well as in the vertebral, tibial and femoral BM (Fig. 4e). In mice with a limited response to immunotherapy (M7 and M10), the activation of iBAT was detected at the peak of immune response, when multiple lymph nodes and the thymus showed substantial increase in [18F]F-AraG signal (Fig. 4d, f). Conversely, therapy-resistant mice exhibited weaker signal in the

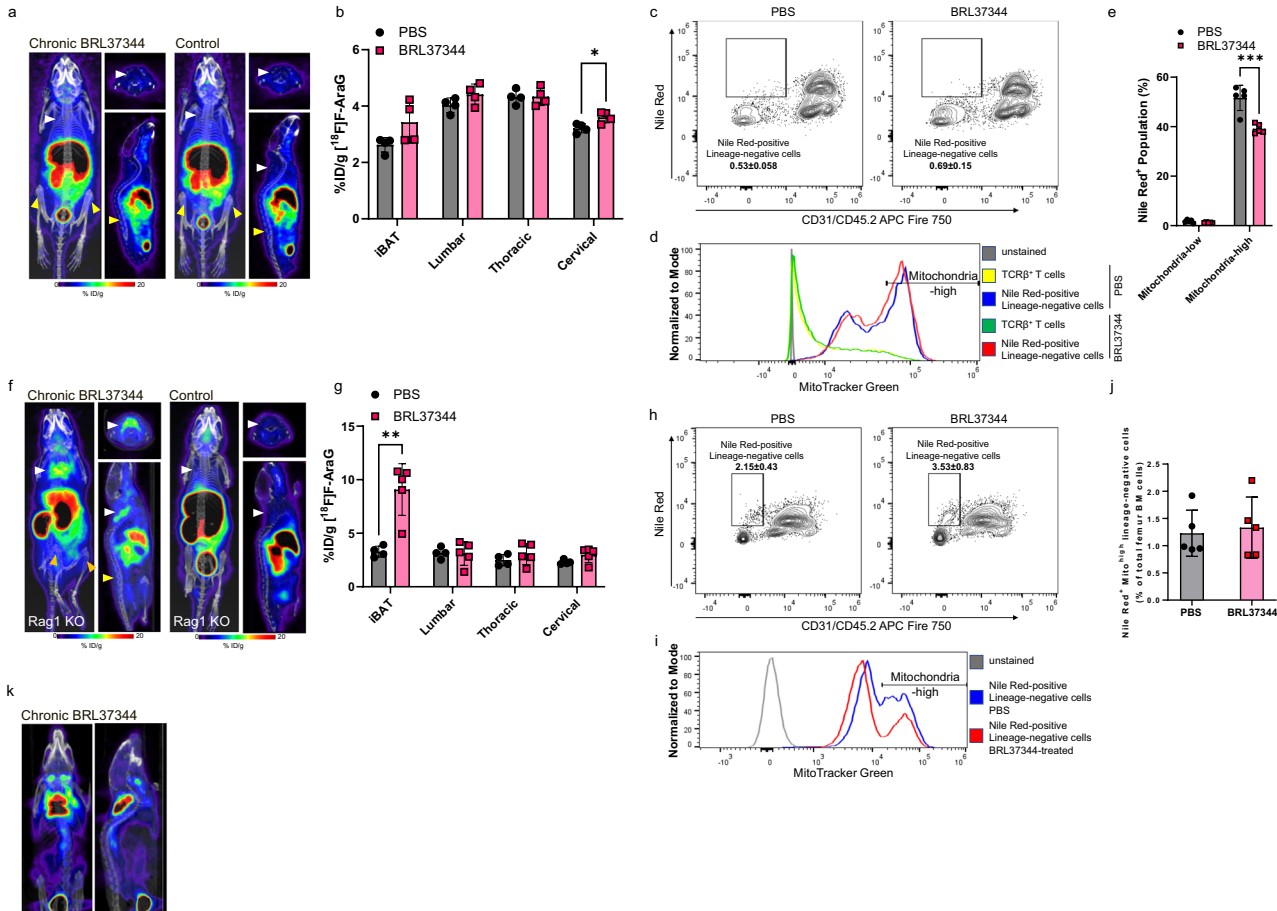

**Fig. 3 | [¹⁸F]F-AraG accumulates in adrenergically stimulated brown and bone marrow adipocytes. a** Chronic adrenergic stimulation of thermogenesis-deficient *Letmd1*-KO mice did not result in increased [¹⁸F]F-AraG uptake in either iBAT or BMAT. **b** BRL37344 treatment did not lead to a significant increase in [¹⁸F]F-AraG accumulation in the iBAT ($p = 0.08$) or lumbar BMAT ($p = 0.14$). Signal in the cervical vertebrae increased marginally (12.5%) post adrenergic stimulation ($p = 0.02$). **c** There was no increase in the abundance of adipocyte or lymphocyte in the femoral bone marrow of adrenergically stimulated Letmd1 KO mice. Frequency of the subset is shown. **d** In *Letmd1*-KO mice, the adipocyte population in the femoral bone marrow constituted the majority of cells with a high mitochondrial content. BRL37344 did not change mitochondrial content of femur bone marrow adipocytes. **e** In femur bone marrow, adrenergic stimulation of *Letmd1*-KO mice decreased the population of adipocytes rich in mitochondria. **f** In T cell-deficient

*Rag1*-KO mice, treatment with BRL37344 resulted in increased accumulation of [¹⁸F]F-AraG in iBAT, whereas there was no notable change in BMAT. **g** After adrenergic stimulation, [¹⁸F]F-AraG signal in the iBAT increased by close to threefold ($p = 0.002$), while the signal in BMAT did not show significant changes. **h** The frequency of adipocytes in the bone marrow of *Rag1*-KO mice increased significantly post adrenergic stimulation ($p = 0.01$). Frequency of the subset is depicted. **i.** *Rag1* deficiency reduced mitochondrial content in femur bone marrow adipocytes. **j** The proportion of mitochondria-rich bone marrow adipocytes remained unchanged in adrenergically stimulated *Rag1*-KO mice ($p = 0.75$). **k** In wt mice, ¹⁸FDG accumulated in the adrenergically stimulated iBAT but not in the BMAT. Data are shown as mean ± SD ($n = 5$). Each spot represents an individual mouse. * $p \leq 0.05$; ** $p \leq 0.01$; *** $p \leq 0.001$; **** $p < 0.0001$.

lymph nodes and no activation of BAT (M9, Fig. 4f). The activation of adipocytes and lymphocytes, as evidenced by the signal in the iBAT and in the lymph nodes, did not persist over time. T cell activity in the lymph nodes displayed a significant reduction following iBAT activation, suggesting immunosuppressive character of activated iBAT (Fig. 4g).

To investigate the connection between neuroinflammation and AT activation, we examined the correlation between [¹⁸F]F-AraG signal in the brain and signal in the iBAT and vertebrae (Fig. 4h, i). In untreated animals, iBAT and signal in the vertebrae correlated with [¹⁸F]F-AraG signal detected in the whole brain as well as with signal at multiple areas in the brain (Fig. 4h). Interestingly, in the treated animals, while the correlations between signals in different brain regions persisted, we did not find any statistically significant correlations between the iBAT and the brain signal (Fig. 4i). Moreover, correlation between the signal in the vertebrae and the brain was largely lost, with the signal in the lumbar vertebrae showing a negative correlation with several areas in the brain. Notable differences between the lumbar signal in treated and untreated

mice were most prominent during the second on-treatment scan (Supplementary Fig. 9b, c).

Collectively these findings demonstrate two key observations: an association between GBM-induced neuroinflammation and activation of AT in certain subjects; and the modulation of this link by immunotherapy.

## Neuroinflammation in multiple sclerosis is associated with activation of BAT and BMAT

Since brain tumors can induce stress response and activate BAT via the SNS independently of the immune system, we examined the relationship between AT activation and neuroinflammation in a mouse model of multiple sclerosis (Fig. 5a). The cuprizone (CPZ) and EAE model allows assessment of two types of brain lesions in the same animal: one that involves innate immune cells, developing at week 3, and the other that implicates T cell infiltration, apparent at week 7. Using this model, we showed previously that [¹⁸F]F-AraG signal in the brain correlates with T cell density[37]. Evaluation of [¹⁸F]F-AraG signal in the iBAT demonstrated that

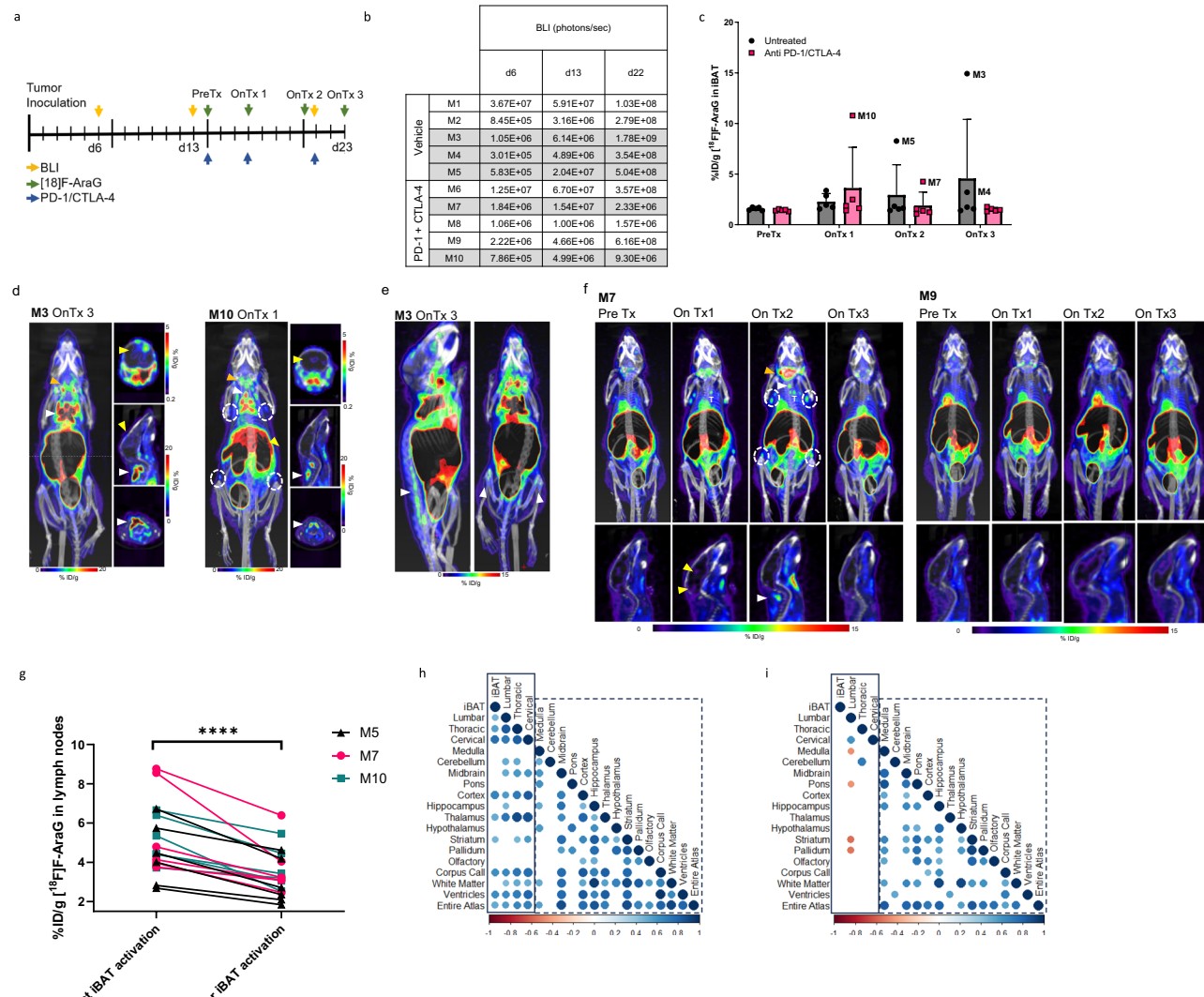

**Fig. 4 | The association of BAT activation with neuroinflammation in GL261 tumors is disrupted by checkpoint inhibitor therapy. a** Mice carrying bioluminescently-tagged GBM tumors were imaged before and during checkpoint inhibitor treatment. Bioluminescence imaging (BLI) was used to track tumor growth. **b** BAT activation was observed in five mice, two treated with anti-PD-1/CTLA-4 antibodies and three untreated mice (gray colored rows). **c** The timing and extent of BAT activation greatly varied between mice. **d** BAT activation was observed in untreated (M3) and mice treated with checkpoint inhibitor therapy (M10) (white arrowheads). The untreated mouse M3 with the most dramatic signal increase in iBAT showed the highest neuroinflammation (yellow arrowhead). Increase in signal in the cervical lymph nodes was observed in both untreated and treated mice (orange arrowheads), while treated mice also showed increase in multiple other nodes in the body, suggestive of systemic response to therapy (dotted circles). **e** In addition to iBAT, high [18F]F-AraG signal was observed in the bone marrow of the lumbar vertebrae, tibia and femur (white arrowheads) of the untreated mouse M3. **f** In a mouse that showed limited response to therapy (M7) signal in the ventricles (yellow arrowhead) and immune activation (lymph nodes, dotted circles, and thymus, T)

was observed early in the therapy. Activation of BAT (white arrowhead) coincided with the peak immune response signified by a large signal increase in the cervical (orange arrowhead), axillary, bronchial and inguinal lymph nodes (dotted circles). In contrast, therapy-resistant mouse M9, showed muted activation in the lymph nodes and no BAT activation. **g** [18F]F-AraG signal in the lymph nodes after iBAT activation was significantly lower compared to the signal at the time of iBAT activation. **h** Correlation matrix of the signal in the iBAT, vertebrae, and brain in untreated mice across all imaging time points. Signal in the iBAT and vertebrae show positive correlation with multiple areas in the brain (full lined rectangle). Dotted rectangle shows correlations between signal in different parts of the brain. **i** Correlation matrix of the signal in the iBAT, vertebrae and brain in treated mice across all imaging time points. The correlations between signal in different brain areas persisted (dotted rectangle), whereas correlations between iBAT, vertebral and signal in the brain significantly weakened, indicating the effect of immunotherapy on the relationship between activated adipose tissue and the brain. Data are shown as mean ± SD. ($n = 5$) Each spot represents an individual animal. **** $p < 0.0001$.

BAT activation coincides with intracerebral T cell infiltration (Fig. 5b–d). While [18F]F-AraG signal in the iBAT did not change significantly after CPZ treatment that leads to microglial/macrophage activation, development of EAE with T cell involvement (Fig. 5d) resulted in a significant increase in iBAT signal (Fig. 5b, c). Increase in the expression of markers of BAT activation was determined in an EAE model (Supplementary Fig. 10a). Furthermore, treatment with fingolimod, a drug that reduces intracerebral lymphocyte infiltration, led to a significant decrease in iBAT signal (Fig. 5b, c).

Signal in the BM of the lumbar vertebrae, tibia and femur, observed in adrenergically stimulated mice and mice with GBM tumors, was also apparent in EAE mice (Fig. 5f). While it is possible that [18F]F-AraG uptake in T cells present in the spinal cord[37] contributes to the detected vertebral signal to a certain extent, close inspection of the signal in the lumbar region strongly suggests that the signal primarily comes from the vertebral body and thus the BM (Supplementary Fig. 10b).

The signals in iBAT and the lumbar region correlated with the brain signal, with correlation for the lumbar vertebrae being considerably stronger

**Fig. 5 | Adipose tissue activation is associated with T cell mediated neuroinflammation in experimental model of multiple sclerosis. a** The cuprizone and EAE model involves treatment with cuprizone for 3 weeks followed by MOG immunization at week 5. Mice were imaged with [18F]F-AraG at baseline, at week 3 (lesions driven by innate immune cells) and at week 7 (lesions with T cell involvement). **b** Activation of BAT (white arrowhead) was observed only in mice with brain lesions with T cell involvement (yellow arrowhead). Mice treated with fingolimod, a drug that prevents intracerebral lymphocyte infiltration, did not show activation of BAT. Upper panels show sagittal slices, bottom panels show transverse slices with iBAT (white arrowhead). **c** [18F]F-AraG signal in the iBAT of CPZ-EAE mice was significantly higher than the baseline ($p = 0.01$), cuprizone treated mice ($p = 0.005$) and CPZ-EAE mice treated with fingolimod ($p = 0.001$). **d** Immunofluorescent CD3 staining confirmed presence of T lymphocytes in the corpus callosum. **e** CPZ-EAE mice showed high [18F]F-AraG signal in the bone marrow of the lumbar vertebrae (upper panel, white arrowhead) and proximal tibia and femur (bottom panel, yellow arrowheads). **f** [18F]F-AraG signal detected in the iBAT showed correlation with the signal observed in the brain. **g** [18F]F-AraG signal in the lumbar vertebrae strongly correlated with the signal in the brain. Data are shown as mean ± SD. ($n = 10$ for baseline, W7 CPZ-EAE and W7 CPZ-EAE + fingolimod, $n = 5$ for W3 CPZ) Each spot represents an individual animal. *$p \leq 0.05$, **$p \leq 0.01$.

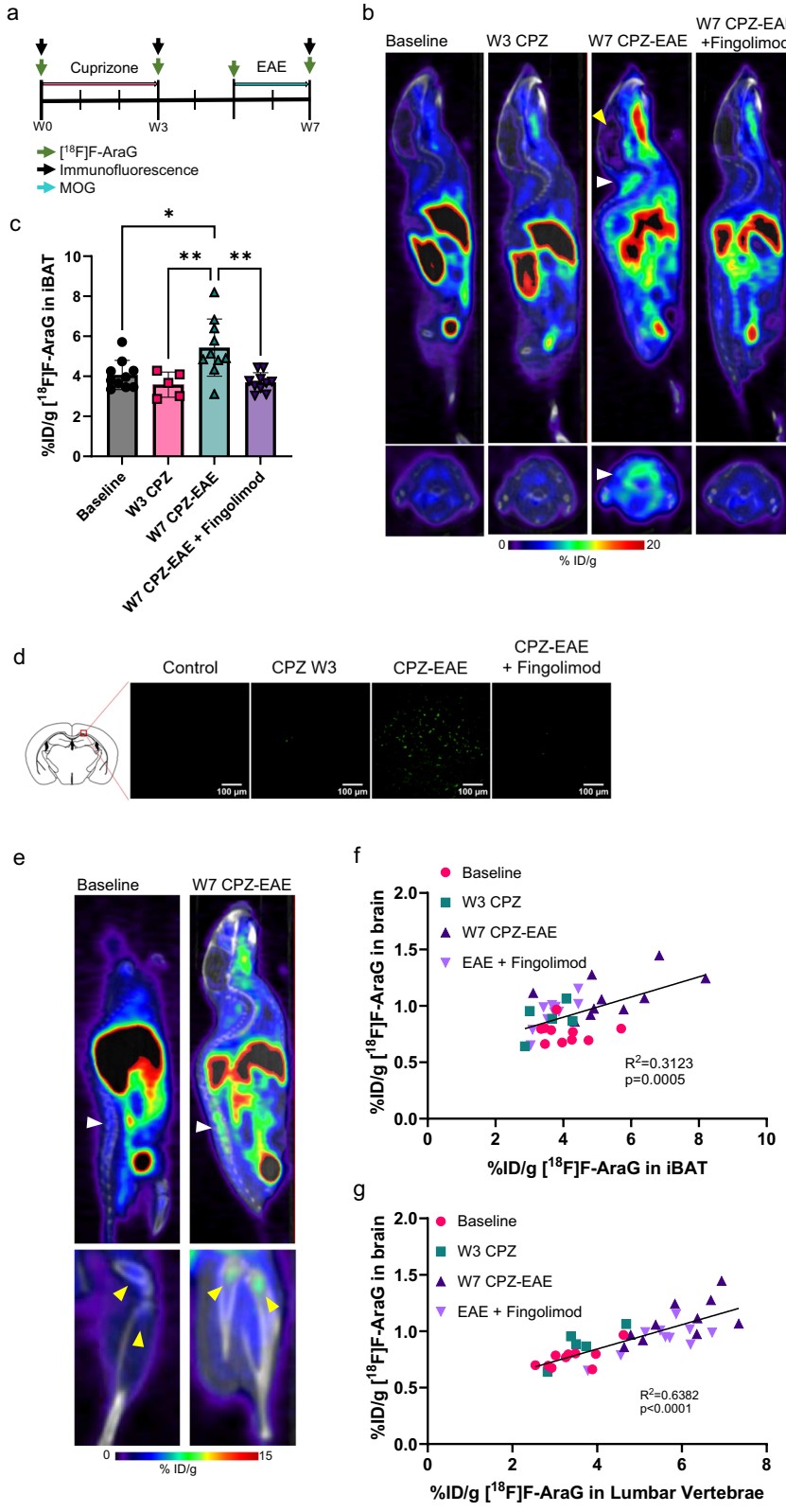

than the one for the iBAT (Fig. 5f, g). As quantitative analysis of iBAT presents a technical challenge[38], the weaker correlation noted for iBAT may be a result of the sampling error related to the quantification method employed.

Overall, these results confirm the existence of a link between neuroinflammation and AT activation, and implicate T cells in the communication between the brain, BAT and BMAT.

## [18F]F-AraG detects activated brown fat in humans

Envisioning the potential impact of immuno-neuro-adipose communication on treatment of diseases like cancer and diabetes, we sought instances of the cooccurrence of neuroinflammation and BAT activation in humans. [18F]F-AraG is currently being investigated in multiple clinical trials, including one examining its distribution in subjects following COVID-19 infection. As the post-acute COVID subjects showed [18F]F-AraG

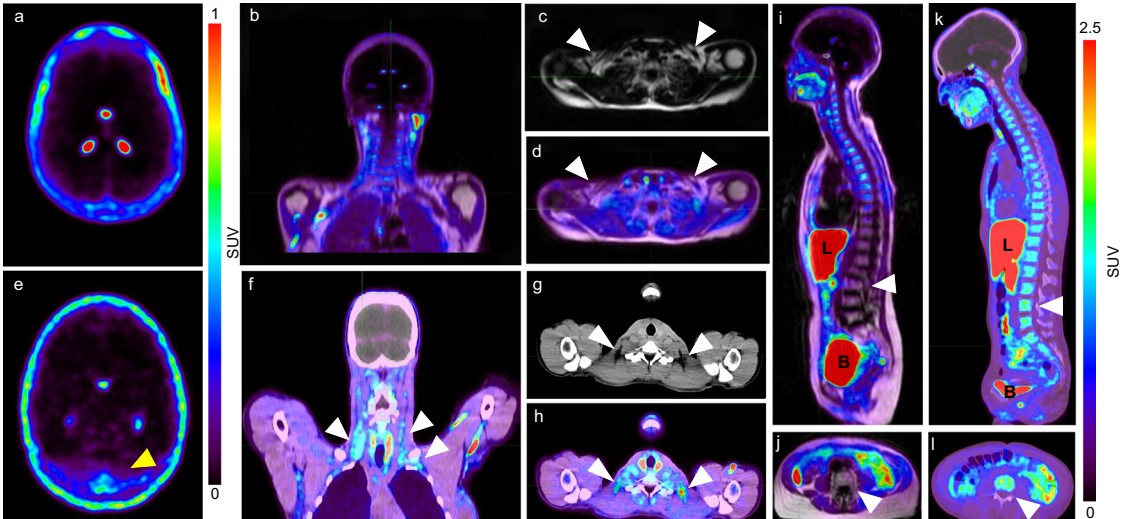

**Fig. 6 | Simultaneous imaging of activated lymphocytes and adipocytes with [¹⁸F]F-AraG reveals concurrent neuroinflammation and BAT activation in a post-acute COVID subject. a–d** [¹⁸F]F-AraG PET/MR images of a COVID-naïve subject. **a** Transverse PET brain slice shows signal in the choroid plexus but no neuroinflammation. **b** Coronal PET/MR slice shows low signal in the supraclavicular brown fat depots. **c** Transverse T1 weighted MR slice shows hypersignal in the supraclavicular fat deposits (white arrowheads). **d** The supraclavicular fat deposits show low [¹⁸F]F-AraG PET signal (white arrowheads). **e–h** [¹⁸F]F-AraG PET/CT images of a post-acute COVID subject. **e** Transverse PET brain slice shows signal in the choroid plexus and a diffuse signal at the confluence of sinuses indicating neuroinflammation (yellow arrowhead). **f** Coronal PET/CT slice shows [¹⁸F]F-AraG uptake in the cervical and supraclavicular brown fat depots (white arrowheads). **g** Transverse CT slice shows hyposignal in the cervical fat deposits (white arrowheads). **h** Cervical fat deposits show high [¹⁸F]F-AraG PET signal (white arrowheads). **i** Sagittal view shows low uptake in the lumbar bone marrow of a COVID-naïve subject (white arrowhead). **j** Transverse slice shows minimal uptake in the L4 vertebrae of a COVID-naïve subject. **k** In the post-acute COVID subject high signal in the lumbar bone marrow is observed (white arrowhead) (**l**). Transverse slice shows high [¹⁸F]F-AraG uptake in the L4 vertebrae of a post-acute COVID subject.

accumulation in the brain suggestive of neuroinflammation[39], we hypothesized that in this study group we will also be able to detect a concurrent activation of BAT. Indeed, in 4 of 19 post-acute COVID subjects (Supplementary Table 3) we observed both neuroinflammation and activated BAT, while neither was detected in a pre-pandemic control (Fig. 6). Interestingly, the most pronounced [¹⁸F]F-AraG signal in the brain was detected at the confluence of sinuses (Fig. 6e), recognized as a neuroimmune interface where antigen presentation and activation of T cells takes place[40]. Signal in the cervical and supraclavicular regions corresponded to sites of the most commonly observed BAT depots[41] (Fig. 6f–h). [¹⁸F]F-AraG uptake in the lumbar vertebrae of post-acute COVID subjects was reported to be significantly higher than that of the pre-pandemic controls[39] (Fig. 6i–l). Considering the recent findings that point to the presence of T cells in the BM of post-COVID patients[42], it is reasonable to speculate that the [¹⁸F]F-AraG lumbar signal may result from the tracer's accumulation in both activated lymphocytes and adipocytes.

Overall, these results provide evidence for the co-occurrence of neuroinflammation and AT activation in humans, and show utility of [¹⁸F]F-AraG to simultaneously detect both.

## Discussion

Bidirectional communication between the immune and nervous systems and its importance in maintaining homeostasis are evident in numerous physiological and pathological phenomena. Although immunologically unique, the brain is no longer considered to be immunologically privileged[43]. A diverse population of immune cells is now recognized to be essential for healthy brain development[44] as well as involved in different disorders[45]. Reciprocally, the brain controls immune responses through various regulatory pathways, including by the SNS[46]. All primary and secondary lymphoid organs, as well as almost every tissue in the body, are enervated by sympathetic fibers[47,48]. The effects of norepinephrine, the main neurotransmitter released during SNS activation, on immune function has been extensively studied in both lymphoid and non-lymphoid tissue[49–51]. However, involvement of SNS-responsive non-lymphoid tissue, specifically AT,

in the context of neuro-immuno communication has remained a largely unexplored area. In this study, we present data that establish a close relationship between SNS-reactive BAT and BMAT and neuroinflammation.

Activation of BAT, a densely innervated tissue, is primarily driven by the brain and release of norepinephrine through sympathetic fibers. We mimicked SNS activation of BAT by using a β3 adrenergic agonist to show that [¹⁸F]F-AraG, a PET agent originally developed for imaging activated T cells, also selectively accumulates in adrenergically activated BAT. [¹⁸F]F-AraG is a nucleoside analog phosphorylated primarily by a mitochondrial kinase that supplies nucleotides for mitochondrial DNA synthesis (mtDNA)[52]. Its ability to track activated T cells stems from this association with mitochondrial biogenesis[14]. Upon stimulation, T cells undergo metabolic reprogramming and dramatically increase both mitochondrial mass and mtDNA[53,54], resulting in an increased demand for nucleotides and higher uptake of [¹⁸F]F-AraG (Supplementary Fig. 3). Given the increase in mitochondrial biogenesis in brown adipocytes during activation[55,56], we hypothesized that [¹⁸F]F-AraG could not only visualize activated T cells, but serve as a tool for imaging adipocyte activation. Imaging of thermogenesis- and T cell-deficient mice, established activated brown adipocytes as the primary source of [¹⁸F]F-AraG uptake in iBAT (Supplementary Table 4). Unexpectedly, during longitudinal imaging of GBM-bearing mice with [¹⁸F]F-AraG we observed activation of iBAT coinciding with immune activation and neuroinflammation. Interestingly, correlations found between signal in the iBAT and various areas in the brain in treatment-naïve mice were absent in mice treated with immunotherapy, indicating the effect of therapy on the BAT-neuroinflammation relationship. Considering that BAT expresses the checkpoint inhibitors targeted by immunotherapies[57], it is conceivable that the therapy might be directly affecting adipocytes. However, it is more plausible that the observed disruption of the correlation stems from the effects of checkpoint inhibitors on their intended target - T cells. It is reasonable to expect that immunotherapy-induced changes in T cell function that improve antitumor capacity, also alter interactions with other tissues, especially immunologically relevant ones, like BAT. Imaging of the cuprizone-EAE model confirmed correlation of BAT activation with

neuroinflammation, and indicated T cell involvement in immuno-neuro-BAT communication.

BMAT is also under the control of the central nervous system[58] but relatively little is known about its function and metabolism. BMAT shares similarities with both white and brown AT[59], yet remains distinct from both[3,34,60]. In mice, two types of BMAT have been identified: rBMAT, responsive to cold and adrenergic stimuli, and constitutive BMAT, resistant to these inducements[26,34]. To date PET imaging of metabolically active BMAT has not been reported[3]. However, [18F]F-AraG imaging of chronically stimulated mice, revealed increased signal at skeletal sites that closely matched rBMAT locations. Consistent with the known inhibitory effects of β-adrenergic signaling on immune function[61], adrenergic stimulation led to a significant reduction in mitochondrial content of BM T cells, indicating a state of metabolic insufficiency[29,62]. Additionally, a higher frequency of the CD69+ and PD-1+ population within the CD4+ subset suggested tissue retention and immune suppression[27,30]. These results are particularly interesting in the context of a study that described sequestration of T cells in the BM when intracranial tumors are present[63]. In line with our observations, this study found that sequestered T cells displayed a loss of sphingosine-1-phosphate receptor 1, a master regulator of lymphocyte egress[64] directly downregulated by CD69[65]. In future studies, it will be prudent to investigate adrenergic signaling as a factor driving lymphodeficiency commonly observed in intracranial tumors. Our preliminary data indicate that the modulation of adrenergic signaling affects lymphocyte infiltration into the GBM tumors (Supplementary Fig. 11). As adrenergic receptors play a role not only in adipocytes, but in immune and cancer cells[66], further research is needed to understand methods to manipulate these different cellular compartments for a maximum therapeutic effect.

In contrast to T cells, adrenergic stimulation resulted in mitochondria-rich adipocyte population almost doubling in size in the BM. Considering [18F]F-AraG's known association with mitochondrial biogenesis[14,18,22], these findings strongly imply that, within the BM, it is the stimulated adipocytes, rather than lymphocytes, that primarily take up the tracer. This is substantiated by the absence of [18F]F-AraG signal in the BM of adrenergically stimulated Letmd-1-KO mice and the corresponding decrease in the number of mitochondria-rich adipocytes. However, T cell presence appears to be crucial for BMAT's stimulation, as indicated by the low [18F]F-AraG accumulation in the BM of adrenergically stimulated Rag1-KO mice.

Interestingly, an increase in [18F]F-AraG's signal in the BM was observed in both GBM and MS models, implying potential involvement of adrenergic signaling in these settings. In immunotherapy treated GBM affected mice, the correlation between the lumbar and brain signal was substantially weakened and even exhibited an inverse association. Given our findings on the role of T cells in BMAT stimulation, we speculate that immunotherapy may modulate T cell-adipocyte communication, resulting in lower BMAT stimulation and increased neuroinflammation. Considering the relevance of immunotherapies in cancer treatment and the propensity of multiple cancer types to metastasize to the BM[67], this hypothesis will be explored in future studies.

Activated BAT has been documented in healthy subjects and cancer patients[68,69], but its connection with neuroinflammation has not been reported to date. Herein, we provide the first evidence for the co-occurrence of neuroinflammation and activated BAT in post-acute COVID subjects, and posit a connection between the two. Post-acute COVID subjects showed elevated [18F]F-AraG signal at the confluence of sinuses, a particularly significant finding given that dural sinuses represent T cells' entry point into the brain parenchyma[40]. The increased signal in the lumbar BM could partly be due to T cell presence in the BM[42], but it likely signifies increased sympathetic activity. The heightened SNS activity was also evident in the [18F]F-AraG accumulation in the cervical and supraclavicular brown fat depots.

This study did not address whether AT activation leads to immunosuppression and reduced neuroinflammation, or if it triggers immune activation and increased intracerebral lymphocyte infiltration. The studies by Jankovic et al. convincingly demonstrated immunosuppressive effects of BAT[10,12]. Another study also showed an anti-inflammatory quality of BAT[9], while a recent report indicated that activation of BAT enhances antitumor response[70]. Although our finding of reduced immune activity in lymph nodes following BAT activation implies immunosuppression, further investigations are needed to definitively characterize the nature of immunomodulation by activated AT. We will also explore whether activated BAT and BMAT act synergistically or antagonistically to one another to modulate neuroinflammation.

The mechanisms operating in neuro-immuno-AT circuitry have not been investigated in this study. However, one can hypothesize that intracerebral immune infiltration may trigger neuronal activation of AT, which in turn, modulates immune responses to lower potentially damaging neuroinflammation. Increasingly appreciated as distinct endocrine organs with an expanding portfolio of batokines, brown and brite AT secrete signaling molecules that regulate inflammatory responses[8] and impact various organs, including the brain[71]. Our future research will focus on identifying signaling factors and networks that enable crosstalk between immune cells, the brain, and AT. Understanding the interconnectedness in this newly proposed circuitry could have implications across a range of research fields, spanning from cancer to neurodegenerative and metabolic diseases. Such comprehension may pave the way for advancements in the treatment and management of these conditions.

## Methods
### Preclinical models
The experiments were conducted at three institutions: University of California, San Francisco (UCSF) (IACUC protocol #AN198359-00B and #AN203775-00) Dana-Farber Cancer Institute, Boston (IACUC protocol #08-023), and University of California, Davis (IACUC protocol number #22889). All animal research was approved by the Animal Care and Use Committee of the corresponding institution. We have complied with all relevant ethical regulations for animal use.

**Adrenergic stimulation**. C57BL/6J female wild type (wt) mice (000664) and C57BL/6J female Rag1 knockout (KO) mice (034159) were purchased from the Jackson Laboratories (6–9-week-old, Bar Harbor, ME). Letmd1 KO mice were generated using CRISPR-Cas9 technology in combination with mouse oviduct electroporation[33]. For the acute adrenergic stimulation, C57BL/6 mice were intraperitoneally (ip) treated with 10 mg/kg dose of β3 agonist BRL37344 (Tocris Bioscience, Minneapolis, MN) 1 h prior to [18F]F-AraG injection and PET/CT imaging. For chronic adrenergic stimulation, mice were injected ip with BRL37344 (10 mg/kg) once a day for 3 days. On the fourth day BRL37344 was administered 1 h prior to [18F]F-AraG injection and PET/CT imaging.

**GBM model**. GBM mouse model was created at Dana-Farber Cancer Institute according to a published procedure[72]. Briefly, luciferase-transduced GL261 cells (GL261-luc2) ($1 \times 10^5$ cells in 2 μL PBS, Perkin-Elmer, Shelton, CT) were injected into the right striatum of anesthetized 6–10-week-old female albino C57BL/6 mice using a Hamilton syringe and stereotactic frame. At day 6, 13, and 22 post tumor implantation, anesthetized mice were injected subcutaneously with D-luciferin at 75 mg/kg (Promega), and imaged with the IVIS Imaging System (Caliper Life Sciences) for 10–120 s. To quantify bioluminescence, identical circular regions of interest were drawn to encircle the entire head of each animal, and the integrated flux of photons (photons per second) in each region of interest was determined using the Living Images software package (Caliper Life Sciences).

**MS model**. MS mouse model was created at the UCSF using a published procedure[37]. Briefly, the cuprizone-EAE model was induced by feeding 8-week-old C57BL/6J female mice a cuprizone supplemented diet (0.25%, Sigma, St. Louis, MO) for 3 weeks, followed by 2 weeks of normal chow. At the start of the sixth week, the mice were immunized with myelin oligodendrocyte glycoprotein (MOG$_{35-55}$) in complete Freund's

adjuvant. The mice received an ip injection of pertussis toxin on the day of immunization and the following day (Hooke Laboratories, Lawrence, MA).

## PET/CT imaging

[18F]F-AraG synthesis. [18F]F-AraG was prepared by UCSF Radiochemistry facility, Dana-Farber Cancer Institute's Molecular Cancer Imaging Facility, and Optimal Tracers (Sacramento, CA) according to the approved IND Chemistry Manufacturing and Control procedures (IND 123591) previously described[17].

Adrenergic stimulation. PET/CT imaging of adrenergic stimulation of wild type and *Rag1* KO mice was performed at UCSF. Imaging of wt mice was performed on an Inveon small animal PET/CT scanner (Siemens Healthcare, Malvern, PA, USA) and imaging of *Rag1* KO mice was performed on nanoScan microPET/CT (Mediso, Budapest, Hungary). Approximately 1 h post BRL37344 treatment, [18F]-FAraG (~7.4 MBq/mouse) was administered intravenously by tail vein injection under anesthesia. Following 18F-FAraG injection, mice were recovered from anesthesia and uptake time of 60 min allowed before the start of the scan. Whole body static scans (15 min PET acquisition followed by CT scan for anatomic reference) were acquired with warming and constant monitoring. Region-of-interest (ROI) analysis of the PET/CT data was performed using VivoQuant software (Invicro, Boston MA). ROIs in the brain were delineated using VivoQuant's mouse brain atlas and the spinal segmentation performed according to the published procedure[73]. ROI in the iBAT was defined by placing a fixed spherical volume in an anatomically accurate location using CT as a guide. Partial volume correction was not performed. The percentage of injected dose per gram was calculated for each ROI. PET/CT imaging of Letmd1 KO mice was performed at UC Davis in a similar fashion.

GBM model. Growth of GL261-luc2 tumors was followed using bioluminescence imaging on day 6 and 13 post tumor implantation. On day 14 tumor-bearing mice were randomized into control and treatment cohorts and imaged before the start of therapy. On the first day of treatment mice were treated with 500 µg of therapeutic (α-PD-1/ α-CTLA-4) or isotype control antibodies, followed by 250 µg dose on day 17 and 22. PET/CT imaging was performed on days 14, 17, 21, and 23 post tumor cell inoculation at the Lurie Family Imaging Center of Dana-Farber Cancer Institute on a Siemens Inveon microPET/CT scanner. Whole body static scans (10 min PET acquisition followed by CT scan for anatomic reference) were acquired ~1 h after [18F]FAraG injection (~6.7 MBq/mouse). The analysis of PET/CT images was performed following the methodology described for adrenergic stimulation.

MS model. The mice were imaged before the disease induction (baseline), after 3 weeks of cuprizone (W3) and post immunization (W7). Whole body PET/CT static scans were acquired and analyzed using the same protocol as described for adrenergic stimulation.

Human imaging. All human subject studies were conducted under an UCSF IRB and radiation safety committee approved protocols. Informed consent was obtained from all individual participants included in the study. All ethical regulations relevant to human research participants were followed. The COVID-19 study (NCT04815096) and the pre-pandemic study in healthy volunteers (NCT02323893) were described in detail elsewhere[14,39].

## Immunohistochemistry and immunofluorescence

The spinal cord was isolated from euthanized mice according to the published procedure[74] and paraffin embedded. Immunohistochemical staining was performed by the Histology and Biomarker Core at UCSF. Immunofluorescence analysis was performed as previously described[37].

## Cell preparation

iBAT was dissected from the intrascapular region. Individual iBAT tissue was minced thoroughly with scissors in M199 buffer (M199 media (Life Technologies, Grand Island, NY) containing 2% BSA and 2.5 mM glucose) before digestion with 1 mg/mL collagenase D (Sigma) and 20 U/mL DNase I (Sigma) at 37 °C in a shaking incubator for 30 min[75–77]. Digested tissue was filtered through a 150-µm cell strainer. Single cell suspension was centrifuged at $350 \times g$ for 5 min. The buoyant adipocytes were enumerated. The pelleted stromal vascular fraction (SVF) cells were re-suspended in red blood cell lysis buffer (Invitrogen, Eugene, OR) and washed with M199 buffer.

To isolate bone marrow (BM) cells from femur, both femurs were isolated and cleaned, and horizontally bisected with a scalpel. Whole BM was flushed by centrifugation at $3000\,g$ for 1 min. Red blood cells were eliminated by brief incubation with red blood cell lysis buffer. The mixture was diluted with M199 buffer, and centrifuged at $1000\,g$ for 1 min. The pelleted cells were resuspended in M199 buffer.

Primary human CD8 + T cells were purchased from Charles River (Northridge, CA). CD8+ T cells were stimulated with soluble anti-CD28 antibody (2 µg/mL, BioLegend, San Diego, CA) for 3 days on anti-CD3 antibody-coated plates (5 µg/mL, BioLegend). [3H]F-AraG uptake was performed as previously reported[15]. Two million cells were incubated with 1 µCi/mL [3H]F-AraG for 2 h at 37 °C. Cells were washed with PBS and lysed with RIPA buffer. Radioactivity of cell lysates was measured on the Beckman LS6500 (Perkin Elmer, Waltham, MA).

## Flow cytometry

Cells were resuspended in staining buffer (Invitrogen), and incubated with Fc-receptor blocker (BioLegend) before staining with fluorochrome-conjugated antibodies (Table S1). Listing of fluorochrome-conjugated antibodies is given in Supplementary Table 1. MitoTracker Green FM (75 nM, Invitrogen) and Nile Red (100 ng/mL, Invitrogen) dyes were added to stain mitochondria and lipid droplets, respectively. Cells were stained with DAPI (100 ng/mL, Sigma) to exclude dead cells. Stained cell events were acquired on Attune NxT flow cytometer (Invitrogen, Carlsbad, CA). Flow cytometry data were analyzed using FlowJo software (TreeStar, Ashland, OR, USA).

## RNA extraction and quantitative reverse transcription polymerase chain reaction (qRT-PCR)

Total RNA was extracted with the QIAGEN RNeasy plus Universal mini kit (Valencia, CA, http://www.qiagen.com) according to the manufacturer's instructions. Total RNA was reverse transcribed into cDNA using SuperScriptIV VILO Master Mix (Invitrogen) according to the manufacturer's protocol. qPCR was performed using PowerTrack SYBR Green Master Mix (Applied Biosystems, Carlsbad, CA) on an QuantStudio 5 384-well machine (Applied Biosystems). Levels of mRNA expression were normalized to 18S ribosomal RNA levels[32]. PCR primers were chosen from previous reports[33,78] and are listed in Supplementary Table 2. Relative expression levels were calculated as $2 - \Delta\Delta CT$[79].

## Statistical analysis and reproducibility

All data are reported as the mean ± standard deviation (SD). Statistical analyses were performed by using an unpaired Student t test using GraphPad Prism (GraphPad Software, La Jolla, CA, USA). Paired Student t test was used for analysis of lymph node signals at the time of BAT activation and following BAT activation (Fig. 4G). An ordinary One-Way ANOVA with Tukey's post hoc test was used to test statistical significance of the adrenergically stimulated wild type mice and for the MS model. Correlation analyses were performed using the Pearson correlation coefficient. Correlation matrices were generated using the "cor" function in the R statistical computing environment (R Core Team, 2022; https://www.R-project.org/)[80] with the aid of Rstudio (Rstudio Team, 2020; http://www.rstudio.com/). $*P \le 0.05$, $**P \le 0.01$, $***P \le 0.001$, $****P \le 0.0001$.

The reproducibility of the findings was supported through the use of standard protocols, multiple replicates, and independent experiments.

## Data availability

The experimental data and the supplementary video that support the findings of this study are available in Dryad with the identifier: https://datadryad.org/stash/share/8ci6uFGo6jOyUpRF26LogIDL4GyUhMa3Z1yjYFHvMSQ.

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

## Acknowledgements

This study is dedicated to the memory of Aruna Gambhir, whose unwavering commitment to advancing technologies aimed at overcoming cancer continue to inspire. This study was supported by NIH SBIR 75N91019C00012 (J.L.), Merck and Co. Investigator Initiated Research Grant (T.H.) and NIH S10OD034286 (Y.S.). We would like to express our gratitude to Drs. Dragan Djordjijević, Luke Whitesell, and Gordana Leposavić for their thoughtful insights and providing constructive feedback on this manuscript. We would also like to acknowledge imaging help of Dr. Douglas Rowland at the UC Davis Center for Molecular and Genomic Imaging.

## Author contributions

Conceptualization: J.L. Methodology: J.L., H.C., C.G., J.Y., P.G., D.R., S.A., M.C. Investigation: C.G., P.G., J.P., L.H., H.C., M.R., J.B., M.P., T.L.H., S.A., M.D., A.B., Q.N., H.S. Experimental design: J.L., H.C. Supervision: J.L., T.J.H., J.Y., P.G., D.R., Y.S., M.C., H.V., H.C. Writing—original draft: J.L.

Writing—review & editing: J.L., H.C., C.G., T.J.H., J.Y., P.G., D.R., Y.S., M.C., H.V.

## Competing interests

The authors declare the following competing interests: J.L., J.P., L.H., H.C., M.R., and H.C. are or were employed by CellSight Technologies. J.L. holds patents related to [$^{18}$F]F-AraG. All other authors declare no competing interests.
