## [Peer Review File · Communications Biology]

Reviewers' comments:

Reviewer #1 (Remarks to the Author):

In the current study, authors reported that brown fat activation is correlated with infiltrated T cell activation in many mouse models and potentially, human clinical subjects.

The study is good, however, the following points need to be addressed for a better presentation. However it can be improved by following:

Majors

Abstract (Line 33-45):

1. Describe more results in the abstract. Authors state the results starting from line 40 (2/3 of the abstract already passed). More results, less introduction.
2. Introduce the function of 18F-AraG and explain its mechanism, particularly focusing on why it can be used in PET-CT imaging for brown fat and T cells, in either the Introduction or Discussion section.

Methods:

1. Line 427: In the Preclinical Models part, list all of the IACUC approval numbers of all the animal experiments
2. Line 560: Authors state that One-Way ANOVA was used to test statistical significance, but with “which” post hoc test to determine statistical significance (maybe Tukey’s)? State the post hoc test.

Results:

1. I think this is the most important part which needs to be clarified. For sure, [18F]F-AraG can label “brown-fat region” with adrenergic stimulation base on the results here. However It is mentioned in Abstract (Line 39) “.....[18F]F-AraG, a mitochondrial metabolic tracer capable of tracking activated lymphocytes and adipocytes” and contents. My question is can [18F]F-AraG really label adipocytes? [18F]F-AraG can label brown fat region or can label brown adipocytes. it should be separated. One explanation is that [18F]F-AraG may label “T cells within brown fat” but may not be able to label “brown adipocytes within brown fat” when stimulated with adrenergic agonists. If yes. authors should provide more evidence that [18F]F-AraG can be take up by brown adipocytes.
2. Figure 3H, BRL37344 stimulated mice contained 3.53 ± 0.83 bone-marrow adipocytes, compared to control mice with 2.15 ± 0.43 bone-marrow adipocytes. Is that really significant? In Results (Line 203) and Figure legends of 3H (Line 812), authors noted “The frequency of adipocytes in the bone marrow of Rag1-KO mice increased significantly post adrenergic stimulation.” 3.53 ± 0.83 (BRL37344 mice) vs 2.15 ± 0.43 (control) does not look to significant to me.
3. Figure 4A. What is BLI? it should be D-luciferin? Or spell and define the full name of BLI in figure legends and contents with first usage.
4. Figure 5A. What is IHC? Define the full name of IHC.
5. Figure legends of Figure 6 Line 880: hypersignal -> hyposignal? Line 880: hyposignal ->

hypersignal?

Minors

1. Line 38: define the full name of the term “¹⁸F-AraG”
2. Line 66: define the full name of the term “brite”. It should be like brown adipocytes-in-white fat depot.
3. Line 78: define the full name of the term “batokines”.
4. Line 122: original, (Figure 123 1D,G). It should be (Figure 123 1D-G)?

Reviewer #2 (Remarks to the Author):

General comments:

In this study Levi and colleagues used [¹⁸F]F-AraG, a mitochondrial tracer, to image activated immune cells and adipocytes simultaneously. They reported that [¹⁸F]F-AraG accumulates in adrenergically-activated BAT and bone marrow adipocytes in mice. They found CNS inflammation was also associated with BAT activation based on [¹⁸F]F-AraG uptake. Finally they showed that [¹⁸F]F-AraG uptake is also present in CNS and BAT of patients after COVID infections. The authors concluded a potential crosstalk between BAT and neuro-inflammation, even though direct evidence has not been presented here. This is an interesting and timely topic and there is need for novel radioligands that can detect activated BAT in humans.

Specific comments:

- 1) Given that cold exposure is the strongest physiologic stimulus for BAT activation, the authors should provide data on [¹⁸F]F-AraG uptake before and after cold exposure (at least in mice).
- 2) The authors concluded that in their models of glioblastoma and multiple sclerosis, neuro-inflammation correlated with BAT activation. However, this was only based on [¹⁸F]F-AraG. This should also be investigated on a molecular level, at least by analyzing gene expression profiles of typical BAT markers such as UCP-1. Ideally, white fat depots should be analyzed as well if any “browning”/“beiging” might occur in these models. They could compare groups with and without specific antitumor treatment.

Reviewer #3 (Remarks to the Author):

The article under review posits a noteworthy correlation between neuroinflammation and brown adipose tissue (BAT) and bone marrow adipose tissue (BMAT) through the utilization of the nuclear medicine imaging tracer F-18 AraG. This discovery introduces a novel and intriguing perspective to the field.

However, the study exhibits certain deficiencies, notably the absence of a standardized

quantification methodology for all positron emission tomography/computed tomography (PET/CT) images. The methodology lacks elucidation regarding the localization of regions of interest (ROIs) and the precise techniques employed for quantifying data from these images.

Moreover, an observation regarding Figure 6K indicates potential image manipulation, particularly evident in the modification of the image window level in the PET image. This manipulation appears to result in varying brightness levels across different panels within the figure, thereby raising concerns about the consistency and integrity of the presented data.

We thank the reviewers for insightful comments and suggestions and wish to respond on a point-by-point basis. Our responses are written in italics.

Reviewer #1:

In the current study, authors reported that brown fat activation is correlated with infiltrated T cell activation in many mouse models and potentially, human clinical subjects.

The study is good, however, the following points need to be addressed for a better presentation. However it can be improved by following:

Majors

Abstract (Line 33-45):

1. Describe more results in the abstract. Authors state the results starting from line 40 (2/3 of the abstract already passed). More results, less introduction.

The abstract has two introductory sentences provided for context and the rest is results (highlighted). In the abstract we have noted the most significant information and findings of the study:

- *A newfound link between immune responses and adipose tissue*
- *Simultaneous imaging of lymphocytes and adipocytes*
- *Models used*
- *Correlation between neuroinflammation and brown and bone marrow adipose tissue*
- *Indication of the described link in humans*
- *Link between immune and nervous systems through adipose tissue as a intermediary*

Brown and brown-like adipose tissues have attracted significant attention for their role in metabolism and therapeutic potential in diabetes and obesity. Despite compelling evidence of an interplay between adipocytes and lymphocytes, the involvement of these tissues in immune responses remains largely unexplored. This study explicates a newfound connection between neuroinflammation and brown and bone marrow adipose tissue. Leveraging the use of [¹⁸F]F-AraG, a mitochondrial metabolic tracer capable of tracking activated lymphocytes and adipocytes simultaneously, we demonstrate, in models of glioblastoma and multiple sclerosis, the correlation between intracerebral immune infiltration and changes in brown and bone marrow adipose tissue. Significantly, we show initial evidence that a neuroinflammation-adipose tissue link may also exist in humans. This study proposes the concept of an intricate immuno-neuro-adipose circuit, and highlights brown and bone marrow adipose tissue as an intermediary in the communication between the immune and nervous systems.

2. Introduce the function of 18F-AraG and explain its mechanism, particularly focusing on why it can be used in PET-CT imaging for brown fat and T cells, in either the Introduction or Discussion section.

As suggested, description of [¹⁸F]F-AraG's mechanism of uptake and explanation of its ability to image both activated T cells and brown adipocytes was added to the discussion:

“We mimicked SNS activation of BAT by using a b3 adrenergic agonist to show that [¹⁸F]F-AraG, a PET agent originally developed for imaging activated T cells, also selectively accumulates in adrenergically activated BAT. [¹⁸F]F-AraG is a nucleoside analog phosphorylated primarily by a mitochondrial kinase that supplies nucleotides for mitochondrial DNA synthesis (mtDNA) [51]. Its ability to track activated T cells stems from this association with mitochondrial biogenesis [14]. Upon stimulation, T cells undergo metabolic reprogramming and dramatically increase both mitochondrial mass and mtDNA [52, 53], resulting in an increased demand for nucleotides and higher uptake of [¹⁸F]F-AraG. Given the increase in mitochondrial biogenesis in brown adipocytes during activation [54, 55], we hypothesized that [¹⁸F]F-AraG could not only visualize activated T cells, but serve as a tool for imaging adipocyte activation.”

Methods:

1. Line 427: In the Preclinical Models part, list all of the IACUC approval numbers of all the animal experiments.

We added this information to the materials and methods section; preclinical models:

The experiments were conducted at three institutions: University of California, San Francisco (UCSF) (IACUC protocol #AN198359-00B and #AN203775-00) Dana-Farber Cancer Institute, Boston (IACUC protocol #08-023) and University of California, Davis (IACUC protocol number #22889).

2. Line 560: Authors state that One-Way ANOVA was used to test statistical significance, but with “which” post hoc test to determine statistical significance (maybe Tukey's)? State the post hoc test.

We stated the post hoc test used in the statistical analysis section:

An ordinary One-Way ANOVA with Tukey's post hoc test was used to test statistical significance of the adrenergically stimulated wild type mice and for the MS model.

Results:

1. I think this is the most important part which needs to be clarified. For sure, [¹⁸F]F-AraG can label “brown-fat region” with adrenergic stimulation base on the results here. However It is mentioned in Abstract (Line 39) “.....[¹⁸F]F-AraG, a mitochondrial metabolic tracer capable of tracking activated lymphocytes and adipocytes” and contents. My question is can [¹⁸F]F-AraG really label adipocytes? [¹⁸F]F-AraG can label brown fat region or can label brown adipocytes. it should be separated. One explanation is that [¹⁸F]F-AraG may label “T cells within brown fat” but may not be able to label

“brown adipocytes within brown fat” when stimulated with adrenergic agonists. If yes, authors should provide more evidence that [¹⁸F]F-AraG can be taken up by brown adipocytes.

The experiments and data that demonstrated that within BAT [¹⁸F]F-AraG indeed accumulates in brown adipocytes are described in the section subtitled: “[¹⁸F]F-AraG accumulates in adrenergically stimulated brown and bone marrow adipocytes”

*To clearly distinguish [¹⁸F]F-AraG’s uptake in T cells from adipocytes within BAT and BMAT tissue we studied adrenergic stimulation in two mouse models that have only one of those functioning or present: *Letmd-1* KO mice have thermogenesis deficient adipocytes but functioning T cells, while *Rag1* KO model, has functioning adipocytes, but is deficient in T cells. In contrast to the wild type mice, chronic adrenergic stimulation of *Letmd1*-KO mice did not lead to uptake of [¹⁸F]F-AraG in either iBAT or lumbar, tibial and femoral BMAT (Figure 3), indicating that, within these tissues, the tracer accumulates primarily in stimulated adipocytes:*

Adrenergic treatment of T cell-deficient, RAG1 KO, mice resulted in [¹⁸F]F-AraG uptake in iBAT, indicating activated brown adipocytes as the primary target of [¹⁸F]F-AraG accumulation in this tissue:

In addition, ^{18}F FDG that accumulates in activated immune cells and stimulated iBAT, but not in rBMAT, did not show high uptake in the BMAT providing evidence that within BMAT [^{18}F]F-AraG accumulates in rBM adipocytes:

Chronic BRL37344

Ex vivo changes in the tissue supported imaging results.

For easier understanding we added the table that summarizes these findings in the Supplementary information:

	Wild Type (no deficiency in T cells or adipocytes)	Letmd1-KO (deficiency in adipocytes)	Rag1-KO (deficiency in T cells)
[¹⁸F]F-AraG signal in the BAT	+	-	+
[¹⁸F]F-AraG signal in the BMAT	+	-	-
Cellular changes in BAT post adrenergic stimulation	Increase in T cells Increase in brown adipocytes	Decreased mitochondrial content in iBAT adipocytes	No T cells
Cellular changes in BMAT post adrenergic stimulation	No increase in T cells, increase in mitochondria rich adipocytes)	Reduction of mitochondria-rich adipocyte population	No increase in mitochondrial content of adipocytes

2. Figure 3H, BRL37344 stimulated mice contained 3.53±0.83 bone-marrow adipocytes, compared to control mice with 2.15±0.43 bone-marrow adipocytes. Is that really significant? In Results (Line 203) and Figure legends of 3H (Line 812), authors noted “The frequency of adipocytes in the bone marrow of Rag1-KO mice increased significantly post adrenergic stimulation.” 3.53±0.83 (BRL37344 mice) vs 2.15±0.43 (control) does not look to significant to me.

The significance in the cited sentence refers to statistical significance, p=0.011. In the bone marrow, adipocytes were the only cell subset that increased with statistical significance post chronic adrenergic stimulation. Despite this increase, because mitochondria-high population (Figure 3J) remained constant [¹⁸F]F-AraG in BMAT did not increase.

We added p value in the figure legend to indicate statistical significance:

H. *The frequency of adipocytes in the bone marrow of Rag1-KO mice increased significantly post adrenergic stimulation (p = 011).*

3. Figure 4A. What is BLI? it should be D-luciferin? Or spell and define the full name of BLI in figure legends and contents with first usage.

We added the full name in the figure 4 legend:

A. *Mice carrying bioluminescently-tagged GBM tumors were imaged before and during checkpoint inhibitor treatment. Bioluminescence imaging (BLI) was used to track tumor growth.*

4. Figure 5A. What is IHC? Define the full name of IHC.

Full name was added in the Figure 5A:

5. Figure legends of Figure 6
 Line 880: hypersignal -> hyposignal?
 Line 880: hyposignal -> hypersignal?

Figure 6 contains both PET/MR (normal subject) and PET/CT images (COVID19 subject). Both MR and CT are used to provide anatomical reference for the PET signal. In MR the fat looks bright, thus the term hypersignal. In CT the fat appears dark, thus the term hyposignal.

Minors

1. Line 38: define the full name of the term “¹⁸F-AraG”

The full name is provided:

The observation of these phenomena was achieved by the use of a mitochondrial metabolic tracer, [¹⁸F]F-AraG (2'-deoxy-2' [¹⁸F]Fluoro-9-β-D-arabinofuranosylguanine) [14], that has a distinctive ability to not only detect mitochondrial changes in activated T cells [15-18], but reveal increased mitochondrial biogenesis in AT as well.

2. Line 66: define the full name of the term “brite”. It should be like brown adipocytes-in-white fat depot.

The full name is provided:

White and brown adipose tissues are the most studied, but brite (brown-in-white) [2] and bone marrow (BM) [3] are gaining increasing interest.

3. Line 78: define the full name of the term “batokines”.

The explanation of what batokines are is provided:

The connection between brown fat and immunity has not been studied extensively, but new studies provide evidence that batokines, signaling molecules secreted by BAT, affect not only metabolism but also systemic immune responses [8, 9].

4. Line 122: original, (Figure 123 1D,G). It should be (Figure 123 1D-G)?

This was corrected:

In comparison to acutely treated mice, chronic adrenergic stimulation led to a comparable [¹⁸F]F-AraG accumulation in iBAT (Supplementary Figure 1A), but significantly increased uptake in the vertebrae, especially in the lumbar region (Figure 1D-G).

Reviewer #2 (Remarks to the Author):

General comments:

In this study Levi and colleagues used [¹⁸F]F-AraG, a mitochondrial tracer, to image activated immune cells and adipocytes simultaneously. They reported that [¹⁸F]F-AraG accumulates in adrenergically-activated BAT and bone marrow adipocytes in mice. They found CNS inflammation was also associated with BAT activation based on [¹⁸F]F-AraG uptake. Finally they showed that [¹⁸F]F-AraG uptake is also present in CNS and BAT of patients after COVID infections. The authors concluded a potential crosstalk between BAT and neuro-inflammation, even though direct evidence has not been presented here. This is an interesting and timely topic and there is need for novel radioligands that can detect activated BAT in humans.

Specific comments:

1) Given that cold exposure is the strongest physiologic stimulus for BAT activation, the authors should provide data on [¹⁸F]F-AraG uptake before and after cold exposure (at least in mice).

We included the results of the cold-exposure experiment in the supplementary material and referred to it in the main text:

Cold exposure also led to tracer accumulation in iBAT (Supplementary Figure 1).

Supplementary Figure 1. [¹⁸F]F-AraG accumulation in iBAT after cold exposure. Mice were kept at 4°C for 24 hours before being imaged with [¹⁸F]F-AraG. (A) Cold exposure leads to [¹⁸F]F-AraG accumulation in iBAT (yellow arrow). Mice kept at room temperature (RT) do not show [¹⁸F]F-AraG signal in the iBAT. (B) [¹⁸F]F-AraG accumulation in iBAT of cold-exposed mice was significantly higher compared to mice maintained at room temperature.

2) The authors concluded that in their models of glioblastoma and multiple sclerosis, neuro-inflammation correlated with BAT activation. However, this was only based on [18F]F-AraG. This should also be investigated on a molecular level, at least by analyzing gene expression profiles of typical BAT markers such as UCP-1. Ideally, white fat depots should be analyzed as well if any “browning”/“beiging” might occur in these models. They could compare groups with and without specific antitumor treatment.

As suggested by the reviewer, to assess brown fat activation on a molecular level we determined expression profiles of the typical BAT activation markers, *Ucp1*, *Dio2* and *Pgc1 α* in EAE model of multiple sclerosis and compared it to unaffected controls. Expression of *Dio2* was significantly increased in EAE mice, while expression of *Ucp1* and *Pgc1 α* showed a trend towards increase but did not reach statistical significance ($p=0.09$). These results agree with varying level of BAT activation observed with [18 F]F-AraG (Figure 5C).

The results are now included in the supplementary material:

Supplementary Figure 10. (A) Relative mRNA expression of thermogenesis-related genes in iBAT of EAE mice. Expression of *Dio2* was significantly increased in EAE mice, while expression of *Ucp1* and *Pgc1 α* showed a trend towards increase but did not reach statistical significance.

We refer to this data in the main text:

Increase in the expression of markers of BAT activation was determined in an EAE model (Supplementary Figure 10A).

Reviewer #3 (Remarks to the Author):

The article under review posits a noteworthy correlation between neuroinflammation and brown adipose tissue (BAT) and bone marrow adipose tissue (BMAT) through the utilization of the nuclear medicine imaging tracer F-18 AraG. This discovery introduces a novel and intriguing perspective to the field.

However, the study exhibits certain deficiencies, notably the absence of a standardized quantification methodology for all positron emission tomography/computed tomography (PET/CT) images. The methodology lacks elucidation regarding the localization of regions of interest (ROIs) and the precise techniques employed for quantifying data from these images.

The quantification methodology that was employed is in accordance with the widely employed and accepted methods that are currently available, and is described in the materials and methods section. To avoid bias we use CT as a guide and place a fixed volume sphere in an anatomically accurate location:

ROIs in the brain were delineated using VivoQuant's mouse brain atlas and the spinal segmentation performed according to the published procedure [72]. ROI in the iBAT was defined by placing a fixed spherical volume in an anatomically accurate location using CT as a guide. Partial volume correction was not performed. The percentage of injected dose per gram was calculated for each ROI.

The limitations of the currently available quantification methods for iBAT are recognized by other researchers in the field and we provided a reference for further reading on that subject in the main text, where we also discuss the effects of these limitations on our results:

As quantitative analysis of iBAT presents a technical challenge [37], the weaker correlation noted for iBAT may be a result of the sampling error related to the quantification method employed.

Moreover, an observation regarding Figure 6K indicates potential image manipulation, particularly evident in the modification of the image window level in the PET image. This manipulation appears to result in varying brightness levels across different panels within the figure, thereby raising concerns about the consistency and integrity of the presented data.

Through the editor we asked for an elaboration of the reviewer's suggestion of image manipulation:

As shown in Figures 6K and 6F, the intensity of the soft-tissue background level is somewhat elevated compared to Figures 6B or 6I, Therefore, there is a possibility that elevated bone marrow activity could be ascribed to increased background intensity.

No image manipulation of any kind was used in Figure 6 (or elsewhere). The images represent print screens of the images as they appear in MIM software, a commonly used software for image visualization. No other filters or image editing was performed. We clearly indicated the scale for PET images so that the images can be properly and accurately compared and contrasted.

The brightness level of the anatomical reference (soft tissue contrast) is different not because we modified the threshold to make certain PET signals appear brighter, but because in the MR (used as anatomical reference for a healthy subject, 6B and 6I) and CT (used as anatomical reference for COVID19 subject, 6F and 6K) all tissues, including fat, appear differently. The differences in [^{18}F]F-AraG uptake in the BAT and bone marrow between the healthy subject and COVID 19 patient remain the same even when only PET images, with no anatomical reference (soft tissue contrast) is shown:

With anatomical reference

Without anatomical reference

REVIEWERS' COMMENTS:

Reviewer #1 (Remarks to the Author):

The authors addressed most of questions.

Reviewer #2 (Remarks to the Author):

All comments have been addressed.

Reviewer #3 (Remarks to the Author):

The authors have addressed the concerns raised in the previous reviews, and the improvements made have significantly enhanced the quality and clarity of the manuscript.

We thank the reviewers for insightful comments and suggestions and wish to respond on a point-by-point basis. Our responses are written in italics.

Reviewer #1:

In the current study, authors reported that brown fat activation is correlated with infiltrated T cell activation in many mouse models and potentially, human clinical subjects.

The study is good, however, the following points need to be addressed for a better presentation. However it can be improved by following:

Majors

Abstract (Line 33-45):

1. Describe more results in the abstract. Authors state the results starting from line 40 (2/3 of the abstract already passed). More results, less introduction.

The abstract has two introductory sentences provided for context and the rest is results (highlighted). In the abstract we have noted the most significant information and findings of the study:

- *A newfound link between immune responses and adipose tissue*
- *Simultaneous imaging of lymphocytes and adipocytes*
- *Models used*
- *Correlation between neuroinflammation and brown and bone marrow adipose tissue*
- *Indication of the described link in humans*
- *Link between immune and nervous systems through adipose tissue as a intermediary*

*Brown and brown-like adipose tissues have attracted significant attention for their role in metabolism and therapeutic potential in diabetes and obesity. Despite compelling evidence of an interplay between adipocytes and lymphocytes, the involvement of these tissues in immune responses remains largely unexplored. **This study explicates a newfound connection between neuroinflammation and brown and bone marrow adipose tissue. Leveraging the use of [¹⁸F]F-AraG, a mitochondrial metabolic tracer capable of tracking activated lymphocytes and adipocytes simultaneously, we demonstrate, in models of glioblastoma and multiple sclerosis, the correlation between intracerebral immune infiltration and changes in brown and bone marrow adipose tissue. Significantly, we show initial evidence that a neuroinflammation-adipose tissue link may also exist in humans. This study proposes the concept of an intricate immuno-neuro-adipose circuit, and highlights brown and bone marrow adipose tissue as an intermediary in the communication between the immune and nervous systems.***

2. Introduce the function of 18F-AraG and explain its mechanism, particularly focusing on why it can be used in PET-CT imaging for brown fat and T cells, in either the Introduction or Discussion section.

As suggested, description of [¹⁸F]F-AraG's mechanism of uptake and explanation of its ability to image both activated T cells and brown adipocytes was added to the discussion:

“We mimicked SNS activation of BAT by using a b3 adrenergic agonist to show that [¹⁸F]F-AraG, a PET agent originally developed for imaging activated T cells, also selectively accumulates in adrenergically activated BAT. [¹⁸F]F-AraG is a nucleoside analog phosphorylated primarily by a mitochondrial kinase that supplies nucleotides for mitochondrial DNA synthesis (mtDNA) [51]. Its ability to track activated T cells stems from this association with mitochondrial biogenesis [14]. Upon stimulation, T cells undergo metabolic reprogramming and dramatically increase both mitochondrial mass and mtDNA [52, 53], resulting in an increased demand for nucleotides and higher uptake of [¹⁸F]F-AraG. Given the increase in mitochondrial biogenesis in brown adipocytes during activation [54, 55], we hypothesized that [¹⁸F]F-AraG could not only visualize activated T cells, but serve as a tool for imaging adipocyte activation.”

Methods:

1. Line 427: In the Preclinical Models part, list all of the IACUC approval numbers of all the animal experiments.

We added this information to the materials and methods section; preclinical models:

The experiments were conducted at three institutions: University of California, San Francisco (UCSF) (IACUC protocol #AN198359-00B and #AN203775-00) Dana-Farber Cancer Institute, Boston (IACUC protocol #08-023) and University of California, Davis (IACUC protocol number #22889).

2. Line 560: Authors state that One-Way ANOVA was used to test statistical significance, but with “which” post hoc test to determine statistical significance (maybe Tukey's)? State the post hoc test.

We stated the post hoc test used in the statistical analysis section:

An ordinary One-Way ANOVA with Tukey's post hoc test was used to test statistical significance of the adrenergically stimulated wild type mice and for the MS model.

Results:

1. I think this is the most important part which needs to be clarified. For sure, [¹⁸F]F-AraG can label “brown-fat region” with adrenergic stimulation base on the results here. However It is mentioned in Abstract (Line 39) “.....[¹⁸F]F-AraG, a mitochondrial metabolic tracer capable of tracking activated lymphocytes and adipocytes” and contents. My question is can [¹⁸F]F-AraG really label adipocytes? [¹⁸F]F-AraG can label brown fat region or can label brown adipocytes. it should be separated. One explanation is that [¹⁸F]F-AraG may label “T cells within brown fat” but may not be able to label

“brown adipocytes within brown fat” when stimulated with adrenergic agonists. If yes, authors should provide more evidence that [¹⁸F]F-AraG can be taken up by brown adipocytes.

The experiments and data that demonstrated that within BAT [¹⁸F]F-AraG indeed accumulates in brown adipocytes are described in the section subtitled: “[¹⁸F]F-AraG accumulates in adrenergically stimulated brown and bone marrow adipocytes”

*To clearly distinguish [¹⁸F]F-AraG’s uptake in T cells from adipocytes within BAT and BMAT tissue we studied adrenergic stimulation in two mouse models that have only one of those functioning or present: *Letmd-1* KO mice have thermogenesis deficient adipocytes but functioning T cells, while *Rag1* KO model, has functioning adipocytes, but is deficient in T cells. In contrast to the wild type mice, chronic adrenergic stimulation of *Letmd1*-KO mice did not lead to uptake of [¹⁸F]F-AraG in either iBAT or lumbar, tibial and femoral BMAT (Figure 3), indicating that, within these tissues, the tracer accumulates primarily in stimulated adipocytes:*

Adrenergic treatment of T cell-deficient, RAG1 KO, mice resulted in [¹⁸F]F-AraG uptake in iBAT, indicating activated brown adipocytes as the primary target of [¹⁸F]F-AraG accumulation in this tissue:

In addition, ^{18}F FDG that accumulates in activated immune cells and stimulated iBAT, but not in rBMAT, did not show high uptake in the BMAT providing evidence that within BMAT [^{18}F]F-AraG accumulates in rBM adipocytes:

Chronic BRL37344

Ex vivo changes in the tissue supported imaging results.

For easier understanding we added the table that summarizes these findings in the Supplementary information:

	Wild Type (no deficiency in T cells or adipocytes)	Letmd1-KO (deficiency in adipocytes)	Rag1-KO (deficiency in T cells)
[¹⁸F]F-AraG signal in the BAT	+	-	+
[¹⁸F]F-AraG signal in the BMAT	+	-	-
Cellular changes in BAT post adrenergic stimulation	Increase in T cells Increase in brown adipocytes	Decreased mitochondrial content in iBAT adipocytes	No T cells
Cellular changes in BMAT post adrenergic stimulation	No increase in T cells, increase in mitochondria rich adipocytes)	Reduction of mitochondria-rich adipocyte population	No increase in mitochondrial content of adipocytes

2. Figure 3H, BRL37344 stimulated mice contained 3.53±0.83 bone-marrow adipocytes, compared to control mice with 2.15±0.43 bone-marrow adipocytes. Is that really significant? In Results (Line 203) and Figure legends of 3H (Line 812), authors noted “The frequency of adipocytes in the bone marrow of Rag1-KO mice increased significantly post adrenergic stimulation.” 3.53±0.83 (BRL37344 mice) vs 2.15±0.43 (control) does not look to significant to me.

The significance in the cited sentence refers to statistical significance, p=0.011. In the bone marrow, adipocytes were the only cell subset that increased with statistical significance post chronic adrenergic stimulation. Despite this increase, because mitochondria-high population (Figure 3J) remained constant [¹⁸F]F-AraG in BMAT did not increase.

We added p value in the figure legend to indicate statistical significance:

H. *The frequency of adipocytes in the bone marrow of Rag1-KO mice increased significantly post adrenergic stimulation (p = 011).*

3. Figure 4A. What is BLI? it should be D-luciferin? Or spell and define the full name of BLI in figure legends and contents with first usage.

We added the full name in the figure 4 legend:

A. *Mice carrying bioluminescently-tagged GBM tumors were imaged before and during checkpoint inhibitor treatment. Bioluminescence imaging (BLI) was used to track tumor growth.*

4. Figure 5A. What is IHC? Define the full name of IHC.

Full name was added in the Figure 5A:

5. Figure legends of Figure 6
 Line 880: hypersignal -> hyposignal?
 Line 880: hyposignal -> hypersignal?

Figure 6 contains both PET/MR (normal subject) and PET/CT images (COVID19 subject). Both MR and CT are used to provide anatomical reference for the PET signal. In MR the fat looks bright, thus the term hypersignal. In CT the fat appears dark, thus the term hyposignal.

Minors

1. Line 38: define the full name of the term “¹⁸F-AraG”

The full name is provided:

The observation of these phenomena was achieved by the use of a mitochondrial metabolic tracer, [¹⁸F]F-AraG (2'-deoxy-2' [¹⁸F]Fluoro-9-β-D-arabinofuranosylguanine) [14], that has a distinctive ability to not only detect mitochondrial changes in activated T cells [15-18], but reveal increased mitochondrial biogenesis in AT as well.

2. Line 66: define the full name of the term “brite”. It should be like brown adipocytes-in-white fat depot.

The full name is provided:

White and brown adipose tissues are the most studied, but brite (brown-in-white) [2] and bone marrow (BM) [3] are gaining increasing interest.

3. Line 78: define the full name of the term “batokines”.

The explanation of what batokines are is provided:

The connection between brown fat and immunity has not been studied extensively, but new studies provide evidence that batokines, signaling molecules secreted by BAT, affect not only metabolism but also systemic immune responses [8, 9].

4. Line 122: original, (Figure 123 1D,G). It should be (Figure 123 1D-G)?

This was corrected:

In comparison to acutely treated mice, chronic adrenergic stimulation led to a comparable [¹⁸F]F-AraG accumulation in iBAT (Supplementary Figure 1A), but significantly increased uptake in the vertebrae, especially in the lumbar region (Figure 1D-G).

Reviewer #2 (Remarks to the Author):

General comments:

In this study Levi and colleagues used [¹⁸F]F-AraG, a mitochondrial tracer, to image activated immune cells and adipocytes simultaneously. They reported that [¹⁸F]F-AraG accumulates in adrenergically-activated BAT and bone marrow adipocytes in mice. They found CNS inflammation was also associated with BAT activation based on [¹⁸F]F-AraG uptake. Finally they showed that [¹⁸F]F-AraG uptake is also present in CNS and BAT of patients after COVID infections. The authors concluded a potential crosstalk between BAT and neuro-inflammation, even though direct evidence has not been presented here. This is an interesting and timely topic and there is need for novel radioligands that can detect activated BAT in humans.

Specific comments:

1) Given that cold exposure is the strongest physiologic stimulus for BAT activation, the authors should provide data on [¹⁸F]F-AraG uptake before and after cold exposure (at least in mice).

We included the results of the cold-exposure experiment in the supplementary material and referred to it in the main text:

Cold exposure also led to tracer accumulation in iBAT (Supplementary Figure 1).

Supplementary Figure 1. [¹⁸F]F-AraG accumulation in iBAT after cold exposure. Mice were kept at 4°C for 24 hours before being imaged with [¹⁸F]F-AraG. (A) Cold exposure leads to [¹⁸F]F-AraG accumulation in iBAT (yellow arrow). Mice kept at room temperature (RT) do not show [¹⁸F]F-AraG signal in the iBAT. (B) [¹⁸F]F-AraG accumulation in iBAT of cold-exposed mice was significantly higher compared to mice maintained at room temperature.

2) The authors concluded that in their models of glioblastoma and multiple sclerosis, neuro-inflammation correlated with BAT activation. However, this was only based on [18F]F-AraG. This should also be investigated on a molecular level, at least by analyzing gene expression profiles of typical BAT markers such as UCP-1. Ideally, white fat depots should be analyzed as well if any “browning”/”beiging” might occur in these models. They could compare groups with and without specific antitumor treatment.

As suggested by the reviewer, to assess brown fat activation on a molecular level we determined expression profiles of the typical BAT activation markers, Ucp1, Dio2 and Pgc1α in EAE model of multiple sclerosis and compared it to unaffected controls. Expression of Dio2 was significantly increased in EAE mice, while expression of Ucp1 and Pgc1α showed a trend towards increase but did not reach statistical significance (p=0.09). These results agree with varying level of BAT activation observed with [¹⁸F]F-AraG (Figure 5C).

The results are now included in the supplementary material:

Supplementary Figure 10. (A) Relative mRNA expression of thermogenesis-related genes in iBAT of EAE mice. Expression of Dio2 was significantly increased in EAE mice, while expression of Ucp1 and Pgc1α showed a trend towards increase but did not reach statistical significance.

We refer to this data in the main text:

Increase in the expression of markers of BAT activation was determined in an EAE model (Supplementary Figure 10A).

Reviewer #3 (Remarks to the Author):

The article under review posits a noteworthy correlation between neuroinflammation and brown adipose tissue (BAT) and bone marrow adipose tissue (BMAT) through the utilization of the nuclear medicine imaging tracer F-18 AraG. This discovery introduces a novel and intriguing perspective to the field.

However, the study exhibits certain deficiencies, notably the absence of a standardized quantification methodology for all positron emission tomography/computed tomography (PET/CT) images. The methodology lacks elucidation regarding the localization of regions of interest (ROIs) and the precise techniques employed for quantifying data from these images.

The quantification methodology that was employed is in accordance with the widely employed and accepted methods that are currently available, and is described in the materials and methods section. To avoid bias we use CT as a guide and place a fixed volume sphere in an anatomically accurate location:

ROIs in the brain were delineated using VivoQuant's mouse brain atlas and the spinal segmentation performed according to the published procedure [72]. ROI in the iBAT was defined by placing a fixed spherical volume in an anatomically accurate location using CT as a guide. Partial volume correction was not performed. The percentage of injected dose per gram was calculated for each ROI.

The limitations of the currently available quantification methods for iBAT are recognized by other researchers in the field and we provided a reference for further reading on that subject in the main text, where we also discuss the effects of these limitations on our results:

As quantitative analysis of iBAT presents a technical challenge [37], the weaker correlation noted for iBAT may be a result of the sampling error related to the quantification method employed.

Moreover, an observation regarding Figure 6K indicates potential image manipulation, particularly evident in the modification of the image window level in the PET image. This manipulation appears to result in varying brightness levels across different panels within the figure, thereby raising concerns about the consistency and integrity of the presented data.

Through the editor we asked for an elaboration of the reviewer's suggestion of image manipulation:

As shown in Figures 6K and 6F, the intensity of the soft-tissue background level is somewhat elevated compared to Figures 6B or 6I, Therefore, there is a possibility that elevated bone marrow activity could be ascribed to increased background intensity.

No image manipulation of any kind was used in Figure 6 (or elsewhere). The images represent print screens of the images as they appear in MIM software, a commonly used software for image visualization. No other filters or image editing was performed. We clearly indicated the scale for PET images so that the images can be properly and accurately compared and contrasted.

The brightness level of the anatomical reference (soft tissue contrast) is different not because we modified the threshold to make certain PET signals appear brighter, but because in the MR (used as anatomical reference for a healthy subject, 6B and 6I) and CT (used as anatomical reference for COVID19 subject, 6F and 6K) all tissues, including fat, appear differently. The differences in [^{18}F]F-AraG uptake in the BAT and bone marrow between the healthy subject and COVID 19 patient remain the same even when only PET images, with no anatomical reference (soft tissue contrast) is shown:

With anatomical reference

Without anatomical reference